# On Time Series Clustering with Graph Neural Networks

**Jonas Berg Hansen**    *jonas.b.hansen@uit.no*
[1] *Department of Mathematics and Statistics, UiT The Arctic University of Norway*

**Andrea Cini**    *andrea.cini@usi.ch*
[2] *Swiss National Science Foundation Postdoc Fellow*
[3] *IDSIA USI-SUPSI, Università della Svizzera italiana*

**Filippo Maria Bianchi**    *filippo.m.bianchi@uit.no*
[1] *Department of Mathematics and Statistics, UiT The Arctic University of Norway*
[4] *NORCE Norwegian Research Centre AS*

**Reviewed on OpenReview:** *https://openreview.net/forum?id=MHQXfiXsr3*

## Abstract

Graph clustering and pooling operators have been adopted in graph-based architectures to capture meaningful patterns in time series data by leveraging both temporal and relational structures. However, the contribution of each design choice and the behavior of different operators remain underexplored. This work introduces a streamlined deep learning framework based on a spatio-temporal graph neural network (STGNN) for clustering time series, which can leverage prior knowledge on the spatial structure of the data. The STGNN-based model flexibly identifies clusters in various data settings through an encoder-decoder architecture with a bottleneck, showing that a spatio-temporal approach can identify meaningful clusters even in datasets that do not explicitly include spatial relations. We validate the framework's qualitative performance through experiments on synthetic and real-world data, showing its effectiveness in different scenarios. We also provide a heuristic for model selection in unsupervised settings via a self-supervised forecasting loss. Code is available at https://github.com/NGMLGroup/Time-Series-Clustering-with-GNNs

## 1 Introduction

In recent years, graph-based time series processing architectures have emerged as powerful tools for handling multivariate time series by modeling with a graph the (spatial) relationships between different sequences (Jin et al., 2024; Cini et al., 2023a). Such relationships can either be given as part of the data or derived from temporal characteristics of the time series (Kipf et al., 2018; Cini et al., 2023c). Clustering and graph pooling have been employed both as preprocessing techniques and as integral components of graph-based architectures to enhance performance in tasks such as forecasting (Cini et al., 2020; 2024) and missing data imputation (Marisca et al., 2024). These methods capture dependencies among time series, enabling the learning of representations that account for multiple scales and dynamics at different resolutions. Despite the effectiveness of clustering and graph pooling components, previous work mainly focused on the performance of downstream tasks and put less emphasis on analyzing the generated clustering partitions. In this sense, a focused study about the capability of graph deep learning models to learn meaningful clusters by integrating both spatial and temporal information is still lacking.

To fill this gap, we analyze a streamlined graph-based model for clustering time series with sparse spatial relations represented by a graph. This model is implemented as a spatio-temporal graph neural network (STGNN) with an encoder-decoder architecture. The STGNN incorporates temporal processing units (Jin et al., 2024), message passing (MP) layers (Battaglia et al., 2018), and a graph clustering (pooling) module (Grattarola et al., 2022). The latter creates a bottleneck positioned between the encoder and the decoder

of the STGNN and learns end-to-end a clustering partition of the time series collection. The pooled representation is then mapped back into the original graph. The full model is trained end-to-end to perform time series forecasting as a (self-supervised) pretext task.

Given the unsupervised nature of the clustering problem, standard cross-validation procedures cannot be followed to select the optimal hyperparameters of the STGNN model. However, we show that, with the proposed encoder-decoder architecture, the forecasting accuracy correlates well with the quality of the extracted clusters. This observation allows for adopting a model selection strategy based on the self-supervised forecasting loss, thus avoiding reliance on supervised information.

To evaluate the effectiveness and limitations of the STGNN model in clustering time series, we perform two types of experiments. The first consists of clustering a set of synthetic data, where both spatial structure and temporal dependencies need to be accounted for to cluster time series correctly. The second is a qualitative analysis of the clusters obtained on a real-world dataset of time series, where the spatial structure is not available as a prior. In the latter case, we show that, even when an explicit graph structure describing the relationships across the time series is not provided, an STGNN can be used to obtain meaningful clusters if the underlying graph is built based on the similarities of the time series.

We summarize our main contributions below.

- We systematically analyze the clustering performance of a reference STGNN model in different settings, both from a quantitative and a qualitative perspective.

- We show that, within our framework, forecasting loss can be used to perform model selection in a completely unsupervised fashion.

- We show that STGNNs can be used to learn meaningful clusters from collections of time series, even when spatial relationships are not given pre-defined.

The paper is structured as follows. Section 2 introduces the background and the problem definition. Section 3 describes the deep learning framework used to perform clustering, along with its training and model selection. In Section 4 we discuss the related work. Section 5 describes the synthetic and real-world datasets. In Section 6, we analyze the performance and the clustering results obtained with the proposed STGNN framework compared to traditional approaches for time series and graph node clustering, deep learning approaches for node clustering of static graphs, and a few variants of the proposed framework. Finally, we draw our conclusions in Section 7.

## 2 Preliminaries

### 2.1 Problem definition

Consider a group of $N$ multivariate time series with $C \geq 1$ variables. We define the observations at time $t$ as $\boldsymbol{X}_t \in \mathbb{R}^{N \times C}$. Furthermore, a sequence of observations from time $t$ to $t+T$ is denoted as $\boldsymbol{X}_{t:t+T} \in \mathbb{R}^{T \times N \times C}$. Analogously, we use $\boldsymbol{x}_t^i \in \mathbb{R}^C$ to denote observations w.r.t. the $i$-th time series at time step $t$. All time series are assumed to be homogenous, i.e., to have the same $C$ channels referring to the same variables. The time series can be associated with $M$ covariates that, at time $t$, are described by a matrix $\boldsymbol{U}_t \in \mathbb{R}^{N \times M}$. We assume that each time series is generated by a time-invariant stochastic process and the presence of Granger causality among them (Granger, 1969).

**Time series clustering** The goal of time series clustering is to group time series that exhibit similar temporal patterns. Standard approaches typically rely on similarity scores among time series or representations thereof. Clustering assignments can be described by a (soft) assignment matrix $\boldsymbol{S} \in \mathbb{R}^{N \times K}$, where $K$ denotes the number of clusters, the entry $s_{i,j}$ is the membership of time series $i$ to cluster $j$, and $\sum_j S_{i,j} = 1$. Hard assignments are obtained by taking the row-wise `argmax` of $\boldsymbol{S}$. The quality of a clustering partition is generally measured in terms of the compactness and separation of the clusters (Maulik & Bandyopadhyay, 2002). Information-theoretic measures can be used to quantify how much the elements in a cluster are similar

to each other and different from those of other clusters (Gokcay & Principe, 2002). Finally, when supervised information is available, the performance can be measured as the agreement between the class and the cluster labels (Rand, 1971). For example, commonly used metrics include normalized mutual information (NMI), homogeneity score (HS), and completeness score (CS). The NMI measures the mutual dependence between the ground-truth labels and the cluster assignments, normalized by their entropy. Formally, it is defined as:

$$\text{NMI}(Y, C) = \frac{2\,I(Y;C)}{H(Y) + H(C)}, \tag{1}$$

where $Y$ denotes the true labels, $C$ the predicted cluster labels, $I(Y;C)$ is the mutual information between $Y$ and $C$, and $H(\cdot)$ represents the entropy. The HS evaluates whether each cluster contains only data points that belong to a single class. It is given by:

$$\text{HS} = 1 - \frac{H(Y|C)}{H(Y)}, \tag{2}$$

where $H(Y|C)$ is the conditional entropy of the class labels given the cluster assignments. The CS assesses whether all data points belonging to the same class are grouped into the same cluster, and is defined as:

$$\text{CS} = 1 - \frac{H(C|Y)}{H(C)}, \tag{3}$$

where $H(C|Y)$ is the conditional entropy of the cluster assignments given the true labels. All these metrics take values in the interval $[0, 1]$, where a value of 1 indicates perfect clustering performance.

**Relational priors** In many real-world scenarios, time series are not independent but show a rich dependency structure, e.g., in sensor networks, traffic data, or biological signals. Specifically, we assume the presence of Granger predictive causality (Granger, 1969) among time series, i.e., we assume that observations at related time series can be used to make more accurate predictions. Further, we assume dependencies among time series to be *sparse* and encode them in an adjacency matrix $\boldsymbol{A} \in \mathbb{R}^{N \times N}$. The entries of $\boldsymbol{A}$ can be binary or real, in which case a weight is associated with each edge. These dependencies, represented as a graph, can be used to inform the clustering process, e.g., as a regularization. Note that we use the term spatial in a broad sense to refer to the dimension spanning the time series collection. In this regard, the adjacency matrix is used to encode functional dependencies and sparsity priors on the dependency structure.

## 2.2 Graph Neural Networks with Pooling

This section introduces the reference graph neural network (GNN) operators used in Section 3 as components of the proposed framework. Hierarchical GNN architectures are usually built by alternating message passing (MP) layers, which use structural relations to update the node representations, with graph pooling layers, which gradually reduce the size of the graph and the node representations (Grattarola et al., 2022).

**Graph convolutional operators** As MP layer, we use as reference a simple graph convolutional operator with a parametrized skip connection (Bianchi et al., 2021). Let $\boldsymbol{H} \in \mathbb{R}^{N \times F}$ be the matrix of the node features associated with the nodes in the graph. Given adjacency matrix $\boldsymbol{A}$, the Graph Conv Skip (GCS) convolutional operator is defined as:

$$\boldsymbol{H}' = \sigma(\boldsymbol{D}^{-1/2}\boldsymbol{A}\boldsymbol{D}^{-1/2}\boldsymbol{H}\boldsymbol{\Theta}_{\text{MP}} + \boldsymbol{H}\boldsymbol{\Theta}_{\text{SC}} + \boldsymbol{b}), \tag{4}$$

where $\boldsymbol{D} = \text{diag}(\boldsymbol{A1})$, and $\boldsymbol{\Theta}_{\text{MP}} \in \mathbb{R}^{F \times F'}$, $\boldsymbol{\Theta}_{\text{SC}} \in \mathbb{R}^{F \times F'}$ and $\boldsymbol{b} \in \mathbb{R}^{F'}$ are learnable parameters.

**Graph pooling** Several families of hierarchical graph pooling operators exist, which differ in how they compute the topology and features of a coarsened graph (Grattarola et al., 2022; Liu et al., 2023a). In this work, we focus on pooling operators that cluster the nodes of the original graph, and then let each cluster become a node in the pooled graph. Specifically, the graph pooling operator learns a soft assignment matrix $\boldsymbol{S} \in \mathbb{R}^{N \times K}$, which maps each node of the original graph into one or more of the $K$ supernodes of the pooled

graph. The assignment matrix $\boldsymbol{S}$ is typically obtained by passing node features learned by one or more MP layers into a learnable function such as a multilayer perceptron (MLP). The structure of $\boldsymbol{S}$ is influenced by the graph topology, the node features, the loss of the downstream task, and a set of auxiliary losses that encourage the formation of well-behaved clusters (Grattarola et al., 2022). The clustering-based pooling operators we consider compute the node features of the pooled graph as:

$$\boldsymbol{H}_{\text{pool}} = \boldsymbol{S}^\top \boldsymbol{H} \in \mathbb{R}^{K \times F}, \tag{5}$$

where the $k$-th row $\boldsymbol{H}_{\text{pool}}[k, :]$ is the combination of node features assigned to supernode $k$.

We also consider a lifting (or unpooling) operation used to map the pooled nodes back to the original node space:

$$\boldsymbol{H}_{\text{lift}} = \boldsymbol{S} f(\boldsymbol{H}_{\text{pool}}) \in \mathbb{R}^{N \times F}, \tag{6}$$

where $f(\cdot)$ represents processing (e.g., the application of one or more MP layers) between the pooling and unpooling operations.

**Graph clustering regularizations** One of the main differences between the existing clustering-based graph pooling methods, such as MinCutPool (Bianchi et al., 2020a), DiffPool (Ying et al., 2018), DMoN (Tsitsulin et al., 2020), and TVGNN (Hansen & Bianchi, 2023), is in how they regularize the learning of the cluster assignment matrix $\boldsymbol{S}$; usually, this happens through the use of an auxiliary clustering loss. The mathematical formulation of the different losses is deferred to Appendix A.

While different methods exist, the loss usually consists of two components (Bianchi, 2022):

$$\mathcal{L}_{\text{aux}} = \mathcal{L}_{\text{topo}} + \mathcal{L}_{\text{quality}}. \tag{7}$$

The first component, $\mathcal{L}_{\text{topo}}$, ensures that the cluster assignments are consistent with the graph topology, while $\mathcal{L}_{\text{quality}}$ encourages clusters to be balanced and well-formed. These auxiliary losses are generally inspired by graph-theoretical properties and are typically formulated based on the static structure of the graph. However, when graph nodes represent time series, it is not immediately clear whether and how strongly these regularizations impact the formation of meaningful temporal clusters. The impact of these regularizations may vary depending on how accurately the adjacency matrix $\boldsymbol{A}$ captures relationships relevant to the clustering task. In this study, we explore whether such regularization contributes meaningfully to the clustering process and investigate under which conditions incorporating these losses provides practical advantages.

## 2.3 Spatio-temporal Graph Neural Networks

STGNNs integrate GNNs and temporal processing units to capture dependencies across both the spatial and temporal dimensions. The core component of an STGNN is a spatio-temporal MP operator (Cini et al., 2023a; Marisca et al., 2024). Let $\boldsymbol{h}_{\leq t}^{i,l}$ be the sequence of representations up to $t$ of the $i$-th node at the $l$-th layer. Following Cini et al. (2023a), a generic spatio-temporal MP operator updates representations at each layer as:

$$\boldsymbol{h}_t^{i,l+1} = \gamma^l \left( \boldsymbol{h}_{\leq t}^{i,l}, \underset{j \in \mathcal{N}(i)}{\text{AGGR}} \, \phi^l \left( \boldsymbol{h}_{\leq t}^{i,l}, \boldsymbol{h}_{\leq t}^{j,l}, a_{ji} \right) \right), \tag{8}$$

where $a_{ji}$ is entry $(i, j)$ of $\boldsymbol{A}$, $\mathcal{N}(i)$ is the neighborhood of the $i$-th node, AGGR is a differentiable and permutation-invariant aggregation function (e.g., sum or mean), and $\gamma^l$, $\phi^l$ are differentiable update and message functions, respectively.

STGNN architectures generally follow two main paradigms depending on the order in which temporal and spatial information are processed (Gao & Ribeiro, 2022; Cini et al., 2023a). The *time-then-space* (TTS) paradigm first encodes temporal information and subsequently processes the resulting node embeddings spatially. This improves computational efficiency, as spatial propagation is performed only with respect to the last time step, thus reducing computational overhead and memory consumption. Conversely, the *time-and-space* (T&S) paradigm interleaves temporal and spatial propagation. While the T&S approach is more flexible, the streamlined architecture of TTS models makes them a valid choice in many scenarios, e.g., when scalability is a requirement.

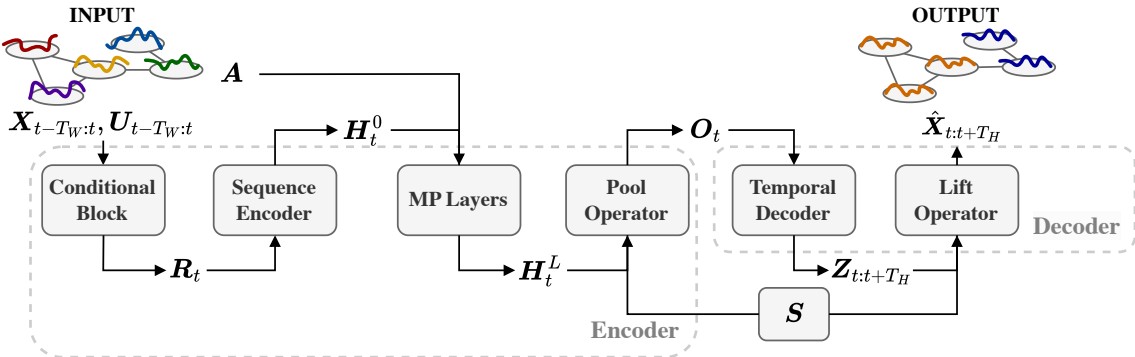

Figure 1: Architectural diagram of the model.

# 3 Time series clustering with STGNNs

In the following, we present a reference architecture for time series clustering with STGNNs. In particular, we propose a time series clustering pipeline where the input time series are first encoded in a static graph representation, which is then pooled into a smaller graph where each node represents a cluster. Representations corresponding to each node are mapped into predictions and then mapped back to the original nodes. In this way, the predictions for each node depend only on their cluster membership. The architecture is trained end-to-end to minimize forecasting errors at each node in a self-supervised fashion, clustering together time series with similar dynamics. The training loss is optimized jointly with a graph clustering loss, which regularizes the learned assignments based on the predefined relational structure. As a result, the graph structure influences the learned model both through the loss function and the message passing operators.

Section 3.1 introduces each processing block, while Section 3.2 discusses the training strategy, the loss function, and model selection.

## 3.1 Model architecture

We start by giving a general overview of the architecture of the STGNN model. Then, we detail the structure of the encoder and decoder. A diagram of the whole architecture is shown in Figure 1.

### 3.1.1 Overview

The proposed architecture is an STGNN with a sequence-to-sequence encoder-decoder structure. The encoder block of the model follows a TTS paradigm, i.e., the temporal dimension is processed first and encoded into a vector embedding for each node. The time embeddings are then processed by a GNN block consisting of stacked MP layers followed by a pooling operator that results in a reduced set of $K < N$ nodes. More details are discussed in Section 3.1.2.

After pooling, the coarsened representation is processed by a decoder that first maps the representation at each node into a different encoding for each target forecasting time step, reintroducing a temporal dimension. We refer to the resulting $K$ time series as *latent factors* (Wang et al., 2019b). Such latent factors can be interpreted as a set of bases, each corresponding to one of the clusters that the model has learned. Since the decoder generates predictions for all the original nodes through a lift operation from such latent factors, their formation directly influences the performance of the downstream task. A detailed description of the decoder is deferred to Section 3.1.3.

### 3.1.2 Encoder

Let $T_W$ be the length of the temporal window of the input time series. At the beginning of the encoder, a combination of two MLPs is used to combine the input time series $\boldsymbol{X}_{t-T_W:t}$ and covariates $\boldsymbol{U}_{t-T_W:t}$ as

$$\boldsymbol{r}_t^i = \text{CondBlock}(\boldsymbol{x}_t^i, \boldsymbol{u}_t^i) \doteq \text{MLP}_X(\boldsymbol{x}_t^i) + \text{MLP}_U(\boldsymbol{u}_t^i), \tag{9}$$

where $\boldsymbol{r}_t^i \in \mathbb{R}^H$, and $H$ is the hidden size, i.e., the embedding dimension. If exogenous variables are absent, $\text{MLP}_U$ is not utilized.

**Sequence encoder**  These initial representations are then processed by a sequence encoder (SeqEnc) which embeds the temporal information of each node into a static vector:

$$\boldsymbol{h}_t^{i,0} = \text{SeqEnc}(\boldsymbol{r}_{T_W:t}^i), \tag{10}$$

where $\boldsymbol{h}_t^{i,0} \in \mathbb{R}^H$; we use $\boldsymbol{H}_t^0 \in \mathbb{R}^{N \times H}$ to denote the matrix obtained by stacking the node representations. The sequence encoder can be implemented by any sequence modeling block, e.g., a temporal convolutional network or recurrent neural network, which produces a vector encoding of the entire sequence.

In practice, as the temporal encoder, we use a stack of dilated temporal convolutional networks (TCNs) (Lea et al., 2016) with increasing dilation factors, to capture at each layer different input dynamics. To obtain a vectorial representation, the $T_W$ representations generated by the TCN layers are combined with an attention-based aggregator.

**Spatial message passing**  After the temporal information has been encoded, the spatial dependencies captured by the graph topology are embedded into node representations through a GNN encoder consisting of a stack of $L$ MP layers:

$$\boldsymbol{H}_t^L = \text{GNN}(\boldsymbol{H}_t^0, \boldsymbol{A}), \tag{11}$$

where $\boldsymbol{H}_t^L \in \mathbb{R}^{N \times H}$ and we use $\text{GNN}(\cdot, \boldsymbol{A})$ to indicate the stack of MP layers applied w.r.t. to $\boldsymbol{A}$. While, in principle, any MP operator can be considered, we use the graph convolution operator of Eq. 4 in our implementation.

**Clustering step**  There is a strong connection between node clustering and the family of graph pooling operators mentioned in Section 2.2, particularly those that optimize losses inspired by graph theoretical objectives, such as graph cuts. Indeed, one can use the graph pooling framework to model the assignment of nodes to supernodes, i.e., clusters. In practice, similarly to Cini et al. (2024), we rely on a static cluster assignment matrix $\boldsymbol{S} \in \mathbb{R}^{N \times K}$ by parameterizing it directly with a table of learnable parameters $\boldsymbol{\Phi} \in \mathbb{R}^{N \times K}$. By doing so, we ensure consistency in the cluster assignments across time steps. Note that this is different from what is typically done in standard pooling operators, where the cluster assignment matrix is obtained as a function of input node features (Grattarola et al., 2022).

To obtain normalized assignments, we apply a row-wise `softmax` to the unnormalized scores $\boldsymbol{\Phi}$. The output of the GNN encoder is then pooled to obtain a reduced node set with $K$ nodes as:

$$\boldsymbol{O}_t = \widetilde{\boldsymbol{S}}^T \boldsymbol{H}_t^L, \quad \widetilde{\boldsymbol{S}}_{i,k} = \frac{\boldsymbol{S}_{i,k}}{\sum_{i=1}^N \boldsymbol{S}_{i,k}}, \tag{12}$$

where $\boldsymbol{O}_t \in \mathbb{R}^{K \times H}$ and $\widetilde{\boldsymbol{S}}$ is the rescaled cluster assignment matrix. By scaling $\boldsymbol{S}$ before pooling, the magnitude of the resulting node representations $\boldsymbol{O}_t$ becomes independent of cluster sizes. This prevents the magnitude of each representation from scaling with the number of nodes it is derived from. This mitigates numerical instabilities for very small clusters and facilitates the formation of latent factors that better capture average patterns in the clusters at time step $t$.

### 3.1.3 Decoder

The decoder first processes the pooled representations with an MLP that maps each node representation into a sequence of length $T_H$:

$$\boldsymbol{Z}_{t:t+T_H} = \text{MLP}_{\text{TD}}(\boldsymbol{O}_t), \quad \boldsymbol{Z}_t \in \mathbb{R}^{K \times C}. \tag{13}$$

The time series $\boldsymbol{Z}_{t:t+T_H}$ are the latent factors, which serve as cluster representatives and are instrumental to the reconstruction of the input time series while encouraging meaningful clusters. As described in Eq. 6, the soft assignments are used to lift the features of the pooled graph back to the input node space:

$$\widehat{\boldsymbol{X}}_\tau = \boldsymbol{S}\boldsymbol{Z}_\tau, \quad \tau = t, \ldots, t + T_H. \tag{14}$$

Here, $T_H$ is the horizon of the time series forecasting used to train the model, and $\widehat{\boldsymbol{X}}_{t:t+T_H}$ are the final predictions. By design, the latent factors $\boldsymbol{Z}_{t:t+T_H}$ act as bases for the forecasts, which are obtained as convex combinations as shown in Eq. 14. In the case of sharp cluster assignments, i.e., when the rows of the cluster assignment matrix $\boldsymbol{S}$ are close to being one-hot vectors, each node is almost exclusively mapped to a single cluster. In this scenario, the lift operation directly maps the corresponding latent factor to all nodes in that cluster, which will more explicitly connect forecasting performance to the groupings made during pooling. Conversely, if the cluster assignments are smooth, different latent factors are combined to build a prediction. Therefore, enforcing sharpness in $\boldsymbol{S}$ can act as a further regularization. We propose a training strategy that encourages each latent factor to serve as a distinctive (and representative) *average* predictor for its corresponding cluster, thereby leading to the formation of more meaningful clusters.

## 3.2 Training

The proposed model is trained in a self-supervised fashion to forecast the next $T_H$ steps by minimizing a mean absolute error (MAE) loss $\mathcal{L}_{pred}$. The optimization procedure will also directly affect the parameters in $\boldsymbol{S}$ and, thus, the formation of the clusters.

**Graph-based regularization**  A key aspect of optimizing GNNs with pooling layers is to promote the formation of informative clusters while avoiding the degenerate solutions that often arise at the start of training. Those typically include assigning all nodes to the same cluster or memberships being near-uniform and uninformative. To this end, we consider auxiliary losses provided by the existing graph pooling operators (Grattarola et al., 2022). These losses are used to modify the learnable parameters $\boldsymbol{\Phi}$ that are used to compute the soft assignments $\boldsymbol{S}$. The balance between the forecasting loss and the auxiliary pooling losses is key to capturing both spatial and temporal information, which, in turn, enables obtaining a meaningful clustering result. To control this balance, the pooling losses are weighted with coefficients:

$$\mathcal{L}_{total} = \mathcal{L}_{pred} + c_1\mathcal{L}_{topo} + c_2\mathcal{L}_{quality}, \tag{15}$$

where $\mathcal{L}_{topo}$ and $\mathcal{L}_{quality}$ are the two terms of the auxiliary loss from Eq. 7 associated with the pooling method of choice. As discussed at the end of Section 3.1.3, our goal is to produce sharp assignments to obtain latent factors that represent each cluster well. To this end, during training, we gradually lower the temperature $\tau$ in the `softmax` used to compute the soft cluster assignments $\boldsymbol{S}$.

## 3.3 Model selection

As there are no universal optimal criteria for assessing clustering performance and measuring clustering quality, model selection largely depends on prior assumptions about the relevant structures in the data and priorities regarding the similarities that one wishes to model. For example, we might want to group time series that exhibit similar dynamics or to identify patterns that appear more often in a certain class. Class labels, however, are not available during training and validation in a completely unsupervised setting. This implies that we cannot leverage metrics such as the NMI, HS, and CS to perform model selection, as these metrics quantify the agreement between identified clusters and class labels.

In our architecture, predictions are obtained directly from the latent factors associated with each cluster. As a result, forecasting accuracy will depend on how well each latent factor represents the dynamics of the time series in the associated cluster. As shown in Section 6, our approach highlights a positive correlation between cluster-class correspondence and forecast performance. Specifically, we show a significant correlation between the forecasting loss and the clustering quality evaluated in terms of NMI and HS. Under the assumption that clusters contain time series with similar patterns and that those patterns characterize, to some extent, the different classes, we can perform model selection by heuristically monitoring the forecasting loss $\mathcal{L}_{pred}$ as a proxy of clustering quality.

## 4 Related Work

There is a vast literature on time series clustering methods based on standard statistical approaches (Liao, 2005; Izakian et al., 2015), machine learning methods (Aghabozorgi et al., 2015; Bianchi et al., 2020b), and deep learning approaches (Ma et al., 2019). Similarly, node clustering is a well-known problem in graph processing research (Schaeffer, 2007) that can be addressed with traditional techniques such as spectral clustering (von Luxburg, 2007), traditional deep learning architectures (Wang et al., 2019a; Pan & Kang, 2021), and modern GNNs (Bianchi et al., 2020a; Tsitsulin et al., 2020; Bianchi, 2022). Methods to cluster time series based on modeling spatio-temporal relationships have traditionally relied on density estimation techniques that leverage various forms of kernel methods (Nakaya & Yano, 2010; Hu et al., 2018; Jiang et al., 2012) and ST-DBSCAN (Birant & Kut, 2007; Gomide et al., 2011). A more recent example is the work of Deb & Karmakar (2023), which proposed to adopt a convex combination of spatial and temporal distance matrices. In this work, we mainly focus on spatio-temporal deep learning models for correlated time series.

**Graph clustering and pooling in spatio-temporal graph neural networks (STGNNs)** Recent advancements in spatio-temporal forecasting have leveraged GNNs to handle complex dependencies in spatial and temporal data (Cini et al., 2023a). Similar to our proposed approach, spatio-temporal processing in STGNNs is typically done by alternating MP with temporal processing layers. Several works also leverage clustering techniques to pre-group data points or enhance model performance (Caggiani et al., 2017; Cini et al., 2020). However, the way clustering is applied in these works differs significantly from our proposed graph pooling strategy. For example, Ji et al. (2023) proposes a framework for traffic flow prediction, which leverages adaptive data augmentation and clustering-based self-supervised tasks to model spatial and temporal heterogeneity to enhance the representation of traffic patterns. To enhance traffic flow forecasting accuracy, Chen et al. (2022) leverages spatial-temporal clustering to group traffic nodes based on recent and periodic traffic states, enabling the STGNN model to capture the spatio-temporal dependencies better. Graph pooling has been increasingly adopted in STGNN architectures to learn richer representations, but mostly focusing on improving the performance on downstream tasks rather than on the quality of the clustering partition. Cini et al. (2024) and Ma et al. (2022) incorporate graph pooling operators, such as MinCutPool (Bianchi et al., 2020a) and DiffPool (Ying et al., 2018), to create hierarchical representations of time series data. Marisca et al. (2024) also extract a hierarchical representation but utilize a non-trainable $k$-MIS pooling operator. Other works adopted more conventional pooling operators like global max pooling (Liu et al., 2023b). To our knowledge, the closest related work to our framework is found in Brenner et al. (2023), which presents a GNN model with an encoder-decoder structure employing graph pooling applied to capacity expansion problems. However, their model is trained to perform reconstruction rather than prediction. In addition, they employ a temporal window in processing, whereas our approach integrates both spatial and temporal dynamics more comprehensively. Also, while Brenner et al. (2023) puts some focus on clustering, their work is still mainly performance-driven and does not perform a thorough analysis of the clustering results. In addition, the cluster assignment matrix for graph pooling is computed directly from the output of a block of GNN layers rather than being learned directly, and, thus, the cluster assignments will vary with time rather than being static.

To summarize, all these works demonstrate the utility of graph pooling in learning multi-scale representations, but limited focus is put on the quality and the interpretability of the clustering results, and the extent to which spatial and temporal information is accounted for in producing the groupings.

## 5 Dataset description

This section presents the datasets considered in our experimental evaluation.

### 5.1 Synthetic datasets

We have designed a series of synthetic datasets to assess the capability of the proposed STGNN architecture to leverage both temporal and spatial dependencies to form clusters. Specifically, each dataset consists of a graph with nodes associated with a time series and a class label. The class of a node is determined both by its role in the graph and by the associated time series, in a proportion that varies across datasets. In other

words, in some datasets, the class of a node depends more on the graph topology, and in others it depends more on the characteristics of the time series.

The graph of each dataset is constructed by connecting subgraphs obtained from the Barabási-Albert (BA) model (Barabási & Albert, 1999; Barabási & Frangos, 2002). The nodes of each subgraph are associated with one of the classes, and the subgraph is constructed using a unique choice of parameters in the BA model. The full graph is obtained by connecting all the subgraphs through a few random edges. The BA model was chosen because, compared to other models such as the stochastic block model, its edge sampling algorithm offers a reasonable amount of flexibility to generate subgraphs that differ in both edge density and structure. For example, one can obtain a range of topologies spanning from more sparsely connected tree-like structures to more dense structures with heavily linked hub nodes. Each subgraph is also guaranteed to be connected (no disjoint regions), meaning that only a few edges between the subgraphs are necessary to ensure that the full graph is connected.

The time series are generated using a graph polynomial vector autoregressive (GPVAR) (Zambon & Alippi, 2022) system with local effects. The procedure is inspired by Cini et al. (2023b), with the difference that we use sinusoids to model the local effects

$$\boldsymbol{H}_t = \sum_{l=1}^{L} \sum_{q=1}^{Q} \Theta_{q,l} \boldsymbol{A}^{l-1} \boldsymbol{X}_{t-q}, \tag{16}$$

$$\boldsymbol{X}_{t+1} = \tan(\boldsymbol{H}_t) + \boldsymbol{a}\sin(2\pi t/\boldsymbol{p} + \boldsymbol{\phi}) + \boldsymbol{\eta}_t, \tag{17}$$

where $\boldsymbol{\Theta} \in \mathbb{R}^{Q \times L}$ are the autoregressive coefficients, $\boldsymbol{a}, \boldsymbol{p}, \boldsymbol{\phi} \in \mathbb{R}^N$ denote the sine wave parameters, and $\boldsymbol{\eta}_t \in \mathcal{N}(\boldsymbol{0}, \sigma^2 \boldsymbol{I})$ denote a vector of $N$ random values generated from a Gaussian distribution centered at zero with the variance $\sigma^2$ being a predefined parameter. To inject the class information within the structure of the time series, all the time series associated with nodes of class $k$ share the set of parameters $a_k$, $p_k$, and $\phi_k$. The intra-class differences arise from the Gaussian noise $\boldsymbol{\eta}_t$ and the structure of $\boldsymbol{A}$, which combines each node with different neighbors.

## 5.2 Smart meter dataset

Real-world collections of time series with node-level labels and associated relational information are rare, which complicates computing quantitative evaluations of the partitions as performance metrics such as NMI rely on supervised information. On the other hand, time series classification datasets are common and, even if the graph topology is not part of the original inputs, an STGNN can still effectively perform time series clustering. In this case, a suitable adjacency matrix must be derived from the data in a way that reflects meaningful relational information across the time series.

Specifically, to test the effectiveness of the proposed STGNN model to perform clustering in a real-world setting, we consider the Commission for Energy Regulation (CER) dataset (Commission for Energy Regulation (CER), 2012). This dataset contains time series of energy load consumption from over 5,000 Irish homes and businesses, which were collected between 2009 and 2010. Each time series is associated with an electricity customer labeled as "residential", "SME" (small/medium enterprise), or "other". Since the "other" class primarily consists of unlabeled data from the other two categories, we focus only on clustering time series from the "residential" and "SME" groups. Furthermore, due to the large amount of missing values in the dataset, we filter out customers with more than 5% missing load values. And finally, to somewhat mitigate the imbalance between "residential" and "SME", we randomly sample 25% of the residential customers. These steps yield a final dataset of 1,541 time series, with a 70/30 split between "residential" and "SME".

Because no explicit relational dependencies are provided in the original data, we construct a graph based on the level of association between the time series. In this instance, the prior spatial information that establishes a dependency structure between the time series is based on the assumption that only series exhibiting a degree of similarity should be considered as related. To build the graph, we first compute the similarities among the time series and then apply a $k$-nearest neighbor ($k = 3$) approach to create a sparse connectivity matrix. Then, we binarize and symmetrize the connectivity matrix, obtaining the final

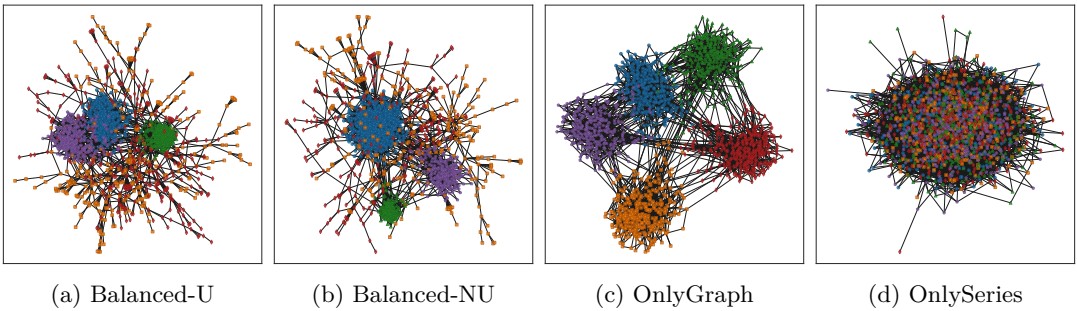

| (a) Balanced-U | (b) Balanced-NU | (c) OnlyGraph | (d) OnlySeries |

Figure 2: Visual representation of the graph of each synthetic dataset. The colors/shapes correspond to the different underlying classes. Node positions were generated using the Fruchterman-Reingold force-directed algorithm.

adjacency matrix $A$ that describes an undirected graph with all edges weighted equally. In Section 6.2, we perform a study to investigate the impact of taking different approaches for building the relational graph, and we evaluate the overall impact of the spatial component of the STGNN. Additional details regarding the CER subset and the constructed graphs are given in Appendix B.2.

## 6 Results and discussion

In this section, we present and discuss the experimental results. Section 6.1 compares the proposed framework against several baselines and provides a qualitative cluster analysis on the synthetic datasets; Section 6.2 investigates the impact of using different approaches to build the adjacency matrix and provides a qualitative cluster analysis on the CER dataset; Section 6.3 reports ablation studies to assess the contribution of individual components; and finally, Section 6.4 evaluates the performance of a few variants of the proposed framework.

For each dataset, a hyperparameter selection was made through a grid sweep where only the pooling loss coefficients were searched, with candidate values consisting of 6 linearly spaced points spanning from 0.1 to 2.5. Furthermore, each configuration was run 3 times, and the configuration that achieved the lowest mean validation mean absolute error (MAE) was selected for evaluation.

We split the datasets along the temporal dimension, where the first 80% of the time series was used for training and the next 10% for validation. The remaining 10% was kept aside for making qualitative plots. For those baselines that did not require hyperparameter selection, we also used the validation set for training.

For the synthetic datasets, the window size $T_W$ was set to 16, and for CER it was set to 72 (hours). For both the synthetic and real-world datasets, the proposed STGNN was set to do a one-step prediction, i.e., $T_H = 1$. Further details about model configuration and additional results are deferred to Appendix C.

### 6.1 Synthetic data clustering

Four different datasets were created following the procedure described in Section 5.1. Each dataset consists of a graph with 1500 nodes divided into 5 classes, and there is a time series of 2000 time steps associated with each node. Fig. 2 shows the graph topology and the class of the nodes in each dataset.

In the first two datasets, named *Balanced-U* and *Balanced-NU*, the parameters of the graph and the time series were chosen such that the class information influences both the spatial and temporal components of the datasets. The only difference between the two datasets is that in Balanced-U the cardinality of the elements in each class is the same, while in Balanced-NU the class distribution is non-uniform. In the latter two datasets, named *OnlyGraph* and *OnlySeries*, the class information characterizes either the graph topology or the time series, respectively, but not both. In the OnlySeries dataset, the underlying graph is a single Erdős-Rényi graph (Erdös & Rényi, 1959) rather than being a composition of different subgraphs, one for each class. On the other hand, in the OnlyGraph dataset, all series are generated from the same AR(2)

model. For OnlyGraph and OnlySeries, the class distribution is uniform. The details about the generations of the different synthetic datasets are reported in Appendix B.1.

As baselines, we compare against methods that either utilize only the spatial or the temporal information. We also consider as baselines architectures for static graphs where the node attributes are a set of temporal and spectral features of the time series. As purely spatial methods, we consider spectral clustering (SC), Walktrap (Pons & Latapy, 2005), and Leiden (Traag et al., 2019), which cluster the nodes by looking at the graph topology. On the other hand, the chosen temporal methods apply spectral clustering to a similarity matrix computed from the time series and disregard the original graph. As time series similarity measures, we use the Euclidean similarity (obtained by taking the inverse of the distance scaled to [0,1]), the cosine similarity, and the dynamic time warping (DTW) algorithm (Berndt & Clifford, 1994). We also consider the cosine similarity between the vectorial embeddings obtained by processing the time series with an echo state network (ESN), as described in Bianchi et al. (2020b).

As discussed in Section 4, there is a lack of STGNN-based approaches specifically designed for clustering time series. To provide a comparison with methods that can consider both time series data and an underlying graph structure, we extract a set of static features from the time series and apply methods for clustering graph nodes that can account for the node attributes. We rely on the TSFEL (Barandas et al., 2020) package to extract the time series features. By using the default parameters, we obtain a total of 165 features consisting of temporal, statistical, and spectral properties of the time series. To process the static graphs, we consider a standard graph autoencoder (GAE) and a variational graph autoencoder (VGAE) (Kipf & Welling, 2016), which are trained to perform link reconstruction. After training, we take the latent representations generated by the encoder of the GAE and VGAE, and we cluster them using a standard $K$-means algorithm. As an additional baseline, we consider a GNN followed by a graph pooling layer, implemented as MinCutPool (Bianchi et al., 2020a), which outputs a soft assignment matrix $\boldsymbol{S}$. This latter method is named *StaticPool* and is trained end-to-end by minimizing the MinCutPool losses. The loss coefficients and softmax temperature used in StaticPool are all fixed at 1. In all these baselines (GAE, VGAE, and StaticPool), the GNN encoder is implemented according to equation 11 and, thus, is similar to the one in our proposed STGNN architecture.

For the proposed STGNN model, we report the results obtained using the auxiliary pooling losses of MinCut-Pool. Also, we report results for a configuration trained only with the forecasting loss named "NoPoolLoss". The results with the other pooling operators are deferred to Appendix C.2. For both the baselines and the proposed STGNN model, the number of clusters $K$ was set equal to the number of node classes.

Numerical results displaying the cluster-class correspondence in terms of NMI, HS, and CS are displayed in Table 1. For each configuration, we report the averages and standard deviations of 5 independent runs. For the balanced datasets, the proposed model with MinCutPool outperforms the other methods in all metrics for both datasets, which shows that the STGNN manages to capture both temporal and spatial information in the data when forming the partition. Additionally, while the NoPoolLoss variant performs reasonably well on average, it fails to leverage the graph information effectively and shows greater inconsistencies in the clustering partitions obtained across the different runs. In the dataset where the class information is only contained in the time series, MinCutPool resulted in more inconsistency than NoPoolLoss. On the other hand, with only graph information, the model with MinCutPool excels, while the NoPoolLoss model fails. It should be noted, however, that in the OnlyGraph dataset, the time series do not contain features that can be leveraged to guide the model selection via the forecasting loss and, thus, the optimal hyperparameter configuration is essentially chosen at random. A more detailed discussion about this point is provided in Appendix C.2.

The performance difference between the proposed model and the methods processing the graphs with static, extracted features can be explained by the main differences in their optimization. GAE, VGAE, and StaticPool are optimized with loss functions that primarily focus on the structural properties of the graph, specifically link reconstruction and graph cut regularization. At the same time, these methods rely on distinctive node attributes to produce meaningful latent representations and subsequent clusters. So, when both the graph and the series have a balanced set of distinctive features, they perform very well. However,

Table 1: Synthetic data experiment results. SC: spectral clustering, DTW: dynamic time warping, ESN: echo state network. For each metric and dataset, the highest value is in bold, and the second highest is underlined. The group *Static feats* refers to the methods processing graphs with static node features extracted from the time series. Values are in percentages (average±standard deviation).

| Method | | Dataset | | | | | |
|---|---|---|---|---|---|---|---|
| | | **Balanced-U** | | | **Balanced-NU** | | |
| | | NMI | HS | CS | NMI | HS | CS |
| **Spatial** | SC | $47.7_{\pm 0.0}$ | $40.2_{\pm 0.0}$ | $58.6_{\pm 0.0}$ | $58.5_{\pm 0.0}$ | $52.6_{\pm 0.0}$ | $65.9_{\pm 0.0}$ |
| | Walktrap | $61.7_{\pm 0.0}$ | $88.0_{\pm 0.0}$ | $47.6_{\pm 0.0}$ | $60.9_{\pm 0.0}$ | $87.7_{\pm 0.0}$ | $46.6_{\pm 0.0}$ |
| | Leiden | $77.7_{\pm 0.0}$ | $91.8_{\pm 0.0}$ | $67.4_{\pm 0.0}$ | $62.8_{\pm 0.0}$ | $86.0_{\pm 0.0}$ | $49.4_{\pm 0.0}$ |
| **Temporal** | Euclidean | $92.3_{\pm 0.0}$ | $92.1_{\pm 0.0}$ | $92.3_{\pm 0.0}$ | $87.2_{\pm 0.0}$ | $88.8_{\pm 0.0}$ | $85.6_{\pm 0.0}$ |
| | Cosine | $91.8_{\pm 0.0}$ | $91.6_{\pm 0.0}$ | $91.9_{\pm 0.0}$ | $86.8_{\pm 0.0}$ | $88.5_{\pm 0.0}$ | $85.3_{\pm 0.0}$ |
| | DTW | $67.0_{\pm 0.1}$ | $63.9_{\pm 0.1}$ | $70.2_{\pm 0.1}$ | $66.4_{\pm 0.4}$ | $65.2_{\pm 0.2}$ | $67.7_{\pm 0.5}$ |
| | ESN | $86.3_{\pm 1.5}$ | $82.4_{\pm 0.7}$ | $90.6_{\pm 2.5}$ | $81.2_{\pm 1.2}$ | $77.8_{\pm 0.9}$ | $84.8_{\pm 1.7}$ |
| **Static feats** | GAE | $94.1_{\pm 6.2}$ | $92.0_{\pm 8.7}$ | $96.4_{\pm 3.3}$ | $93.4_{\pm 5.1}$ | $91.5_{\pm 7.2}$ | $\underline{95.4}_{\pm 3.2}$ |
| | VGAE | $\underline{98.2}_{\pm 0.9}$ | $\underline{98.2}_{\pm 0.9}$ | $\underline{98.2}_{\pm 0.9}$ | $\underline{94.8}_{\pm 2.0}$ | $\underline{94.8}_{\pm 2.0}$ | $94.7_{\pm 2.0}$ |
| | StaticPool | $94.7_{\pm 4.0}$ | $94.5_{\pm 4.2}$ | $94.9_{\pm 3.8}$ | $69.8_{\pm 4.6}$ | $72.2_{\pm 4.9}$ | $67.5_{\pm 4.2}$ |
| **Proposed** | NoPoolLoss | $74.2_{\pm 37.1}$ | $73.9_{\pm 37.0}$ | $94.6_{\pm 2.9}$ | $83.6_{\pm 1.3}$ | $85.0_{\pm 1.5}$ | $82.2_{\pm 1.2}$ |
| | MinCutPool | $\mathbf{100.0}_{\pm 0.0}$ | $\mathbf{100.0}_{\pm 0.0}$ | $\mathbf{100.0}_{\pm 0.0}$ | $\mathbf{95.5}_{\pm 0.2}$ | $\mathbf{95.8}_{\pm 0.2}$ | $\mathbf{95.3}_{\pm 0.2}$ |
| | | **OnlySeries** | | | **OnlyGraph** | | |
| | | NMI | HS | CS | NMI | HS | CS |
| **Spatial** | SC | $0.5_{\pm 0.0}$ | $0.4_{\pm 0.0}$ | $0.6_{\pm 0.0}$ | $\mathbf{100.0}_{\pm 0.0}$ | $\mathbf{100.0}_{\pm 0.0}$ | $\mathbf{100.0}_{\pm 0.0}$ |
| | Walktrap | $7.9_{\pm 0.0}$ | $14.5_{\pm 0.0}$ | $5.4_{\pm 0.0}$ | $85.6_{\pm 0.0}$ | $85.6_{\pm 0.0}$ | $85.6_{\pm 0.0}$ |
| | Leiden | $0.8_{\pm 0.0}$ | $1.1_{\pm 0.0}$ | $0.7_{\pm 0.0}$ | $\underline{99.7}_{\pm 0.0}$ | $\underline{99.7}_{\pm 0.0}$ | $\underline{99.7}_{\pm 0.0}$ |
| **Temporal** | Euclidean | $\mathbf{100.0}_{\pm 0.0}$ | $\mathbf{100.0}_{\pm 0.0}$ | $\mathbf{100.0}_{\pm 0.0}$ | $0.3_{\pm 0.1}$ | $0.3_{\pm 0.1}$ | $0.3_{\pm 0.1}$ |
| | Cosine | $\mathbf{100.0}_{\pm 0.0}$ | $\mathbf{100.0}_{\pm 0.0}$ | $\mathbf{100.0}_{\pm 0.0}$ | $0.3_{\pm 0.0}$ | $0.3_{\pm 0.0}$ | $0.3_{\pm 0.0}$ |
| | DTW | $\underline{99.5}_{\pm 0.0}$ | $\underline{99.5}_{\pm 0.0}$ | $\underline{99.5}_{\pm 0.0}$ | $0.2_{\pm 0.0}$ | $0.2_{\pm 0.0}$ | $0.2_{\pm 0.0}$ |
| | ESN | $99.4_{\pm 0.6}$ | $99.4_{\pm 0.6}$ | $99.4_{\pm 0.6}$ | $0.3_{\pm 0.1}$ | $0.3_{\pm 0.1}$ | $0.3_{\pm 0.1}$ |
| **Static feats** | GAE | $8.7_{\pm 2.3}$ | $8.3_{\pm 2.2}$ | $9.1_{\pm 2.3}$ | $21.5_{\pm 9.8}$ | $20.1_{\pm 9.2}$ | $23.1_{\pm 10.4}$ |
| | VGAE | $11.7_{\pm 0.9}$ | $11.7_{\pm 0.9}$ | $11.8_{\pm 0.9}$ | $41.4_{\pm 8.0}$ | $41.3_{\pm 8.1}$ | $41.5_{\pm 8.0}$ |
| | StaticPool | $13.7_{\pm 2.6}$ | $13.7_{\pm 2.6}$ | $13.7_{\pm 2.6}$ | $6.3_{\pm 1.6}$ | $6.3_{\pm 1.6}$ | $6.3_{\pm 1.6}$ |
| **Proposed** | NoPoolLoss | $\mathbf{100.0}_{\pm 0.0}$ | $\mathbf{100.0}_{\pm 0.0}$ | $\mathbf{100.0}_{\pm 0.0}$ | $0.0_{\pm 0.0}$ | $0.0_{\pm 0.0}$ | $0.0_{\pm 0.0}$ |
| | MinCutPool | $75.1_{\pm 38.6}$ | $74.7_{\pm 38.6}$ | $75.5_{\pm 38.6}$ | $\mathbf{100.0}_{\pm 0.0}$ | $\mathbf{100.0}_{\pm 0.0}$ | $\mathbf{100.0}_{\pm 0.0}$ |

Figure 3: Quantile plot for a snippet of the time series of each cluster for the Balanced-U dataset. The median is given as a solid line, and the corresponding latent factor from the model is given as a dashed line. The shaded areas correspond to the 25% and 75% quantiles.

compared to the proposed approach, they struggle when there is less information either in the spatial or the temporal domain. See Appendix C.2 for additional details and further experiments on this setting.

The latent factors produced by the STGNN model with MinCutPool for a segment of the series are displayed in Fig. 3, together with the quantile plots of the time series in the corresponding clusters. As we can see, the latent factors closely match the cluster medians, and we argue that they can be considered as valid representatives of the clusters.

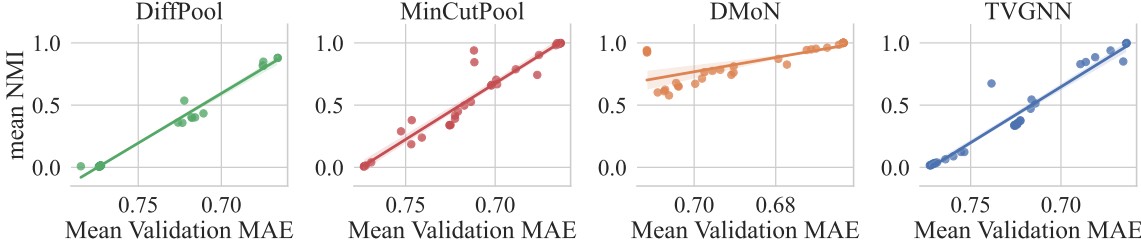

Figure 4: Plot of linear regression for mean NMI as a function of mean validation MAE for the Balanced-U dataset (the x-axis is inverted).

**Unsupervised model selection**    As discussed in Section 3.3, we rely solely on the unsupervised forecasting loss for model selection. Fig. 4 plots the NMI against the MAE for the configurations considered during the hyperparameter sweep of the pooling coefficients. Notably, the configurations that yield a low prediction MAE generally also achieve a high NMI, which supports our hypothesis that the forecasting performance and the quality of the clustering partition are closely related in our proposed architecture. Moreover, this correlation holds even when other losses different from those of MinCutPool are used. We underline that relying on the MAE to perform model selection is a noteworthy result, given the well-known difficulties in tuning the hyperparameters of a clustering model in the absence of supervision.

## 6.2   Qualitative cluster analysis on real-world time series data

The main goal of this experiment is to study the quality of the partitions generated by the proposed spatio-temporal clustering approach when applied to a purely temporal dataset, i.e., where a graph encoding the relationships among the time series is not given. In particular, we consider the real-world time series dataset presented in Section 5.2 and we manually construct an adjacency matrix before applying the given framework.

We consider different methods to construct the adjacency matrix. First, as described in Section 5.2, we build a $k$-nn graph by looking at the similarities between the time series. As the similarities, we consider the Euclidean distance, Pearson correlation, and correntropy (Liu et al., 2007) across the different time series. In addition, we perform an ablation study by considering three adjacency matrices that do not account for the temporal information: an identity matrix, a fully connected matrix, and the adjacency matrix of an Erdős-Rényi graph. These latter configurations represent baselines with non-informative graph structures to explore whether more informative spatial features are beneficial for time series clustering. In each setting, we consider the NoPoolLoss and MinCutPool variants of the proposed framework. Notably, NoPoolLoss, configured with the identity matrix as adjacency, represents a purely temporal deep learning framework optimized solely with a forecasting loss.

We also consider two different settings by letting the number of clusters be either equal ($K = 2$) or greater ($K = 5$) than the number of classes. The latter setting allows for a deeper qualitative assessment of the latent factors as the clusters are smaller, more compact, and should show more diversity. Therefore, in the 5-cluster setting, our quantitative comparisons focus exclusively on the homogeneity score, which measures the class purity within each cluster.

The results comparing the clustering performance relative to the "residential" and "SME" classes are reported in Table 2. In the 2-cluster setting, utilizing an uninformative graph that disregards the relationships among time series results in significantly poor performance. Furthermore, if no pooling losses are applied, the clustering performance does not improve even when more informative graphs are utilized. On the other hand, when the MinCutPool losses are minimized during training, the performance is noticeably enhanced when using an informative graph that describes the relationship between time series. The most significant improvement is observed when the graph is built using the Pearson similarity across the time series. When adding the pooling losses to the configurations with uninformative graphs, the performance did, however, not improve but rather decreased. Additionally, in some of the runs with the model utilizing the fully connected

Table 2: Performance on subsampled CER dataset for the different adjacency construction methods.

| Method | | 2 clusters | | | 5 clusters |
|---|---|---|---|---|---|
| | | NMI | HS | CS | HS |
| **NoPoolLoss** | Identity | $8.1_{\pm2.9}$ | $8.5_{\pm3.0}$ | $7.7_{\pm2.8}$ | $29.3_{\pm3.1}$ |
| | Fully Connected | $8.3_{\pm2.3}$ | $8.7_{\pm2.4}$ | $7.9_{\pm2.3}$ | $28.2_{\pm9.7}$ |
| | Random | $8.4_{\pm2.1}$ | $8.8_{\pm2.2}$ | $8.0_{\pm2.0}$ | $21.1_{\pm11.2}$ |
| | Euclidean | $7.2_{\pm1.3}$ | $7.6_{\pm1.3}$ | $6.9_{\pm1.2}$ | $32.4_{\pm4.2}$ |
| | Pearson | $4.9_{\pm3.1}$ | $5.2_{\pm3.3}$ | $4.7_{\pm3.0}$ | $27.1_{\pm5.7}$ |
| | Correntropy | $5.8_{\pm2.0}$ | $6.1_{\pm2.1}$ | $5.5_{\pm1.9}$ | $29.4_{\pm4.6}$ |
| **MinCutPool** | Identity | $6.9_{\pm2.3}$ | $7.3_{\pm2.5}$ | $6.5_{\pm2.2}$ | $15.1_{\pm9.8}$ |
| | Fully Connected | $0.3_{\pm0.5}$ | $0.2_{\pm0.3}$ | $\mathbf{64.7}_{\pm43.4}$ | $6.3_{\pm10.2}$ |
| | Random | $2.2_{\pm2.6}$ | $2.3_{\pm2.7}$ | $2.1_{\pm2.5}$ | $0.2_{\pm0.2}$ |
| | Euclidean | $\underline{26.8}_{\pm0.9}$ | $\underline{28.0}_{\pm0.9}$ | $25.7_{\pm0.9}$ | $\underline{58.6}_{\pm1.4}$ |
| | Pearson | $\mathbf{45.3}_{\pm2.1}$ | $\mathbf{44.5}_{\pm1.6}$ | $\underline{46.1}_{\pm2.5}$ | $50.3_{\pm1.1}$ |
| | Correntropy | $22.5_{\pm1.1}$ | $23.4_{\pm1.2}$ | $21.7_{\pm1.1}$ | $\mathbf{58.7}_{\pm0.8}$ |

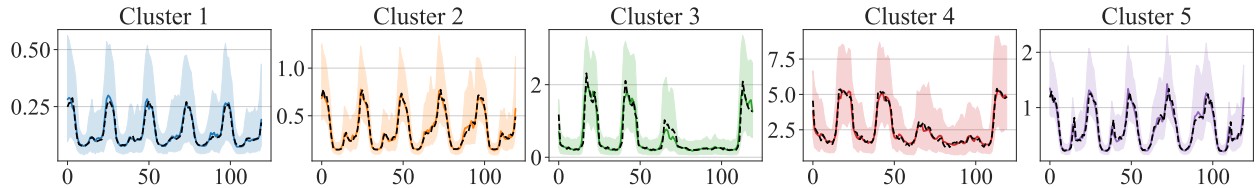

Figure 5: Quantile plot for a portion of the CER test set for the case with 5 clusters obtained with the proposed model applied with the correntropy $k$-nn graph. The cluster median is given as a solid line, 25% and 75% quantiles are given as shaded areas, and the corresponding latent factor is given as a dashed line.

graph, we observed a collapse into a single cluster, which resulted in a high average and standard deviation in CS.

In the 5-cluster scenario, with some exceptions, the different configurations improve in terms of HS. This improvement is expected since a finer partition allows for splitting larger clusters that combine samples from different classes. However, the extent of performance enhancement in terms of HS varies significantly. When the proposed STGNN is configured with MinCutPool, the configurations utilizing Euclidean and correntropy-based graphs exhibit the greatest improvement, with correntropy achieving the highest mean HS value overall. The NoPoolLoss variant also achieves better overall performance when using more clusters and, as expected, outperforms the model with the MinCutPool losses when using the uninformative graphs. This latter result is not surprising because if the graph structure is completely unrelated to the actual time series, optimizing the clustering objective of the MinCutPool losses has an adversarial effect. To further qualitatively analyze the clustering results, in Fig. 5 we compare the latent factors identified by the model using the correntropy-based graph to the quantiles of the time series assigned to each cluster. As we can see, the latent factors closely match the median of the time series distribution within each group, meaning that they can effectively be used as representatives of the different clusters. In Fig. 6 we show the class medians and the class distribution within each cluster. All clusters exhibit a relatively high degree of class purity (i.e., a high HS value), as they mostly contain time series from the same class. This also means that the latent factors closely match the energy consumption profiles for the most frequent class in each cluster. We also note that in clusters like the fifth one, there is a notable resemblance between the median profiles of the two classes.

It is also worth noting that the difference in magnitude between the time series significantly affects the clustering results. This is a direct consequence of how we normalized the data, i.e., we standardized each time series with the global mean and standard deviation computed across the whole dataset. To put more

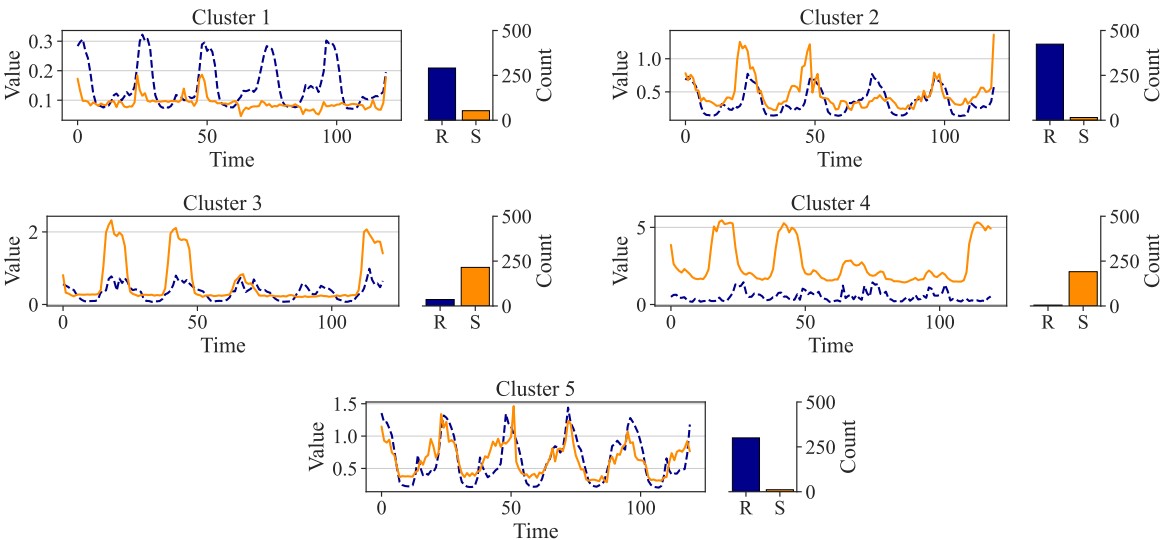

Figure 6: Class distribution within the same clusters as in Fig. 5. For each cluster, the line plot shows the median for each class with "residential" represented as a dashed blue line and "SME" as a solid orange line. The bar plot to the right of each line plot shows the class distribution with R: "residential" and S: "SME".

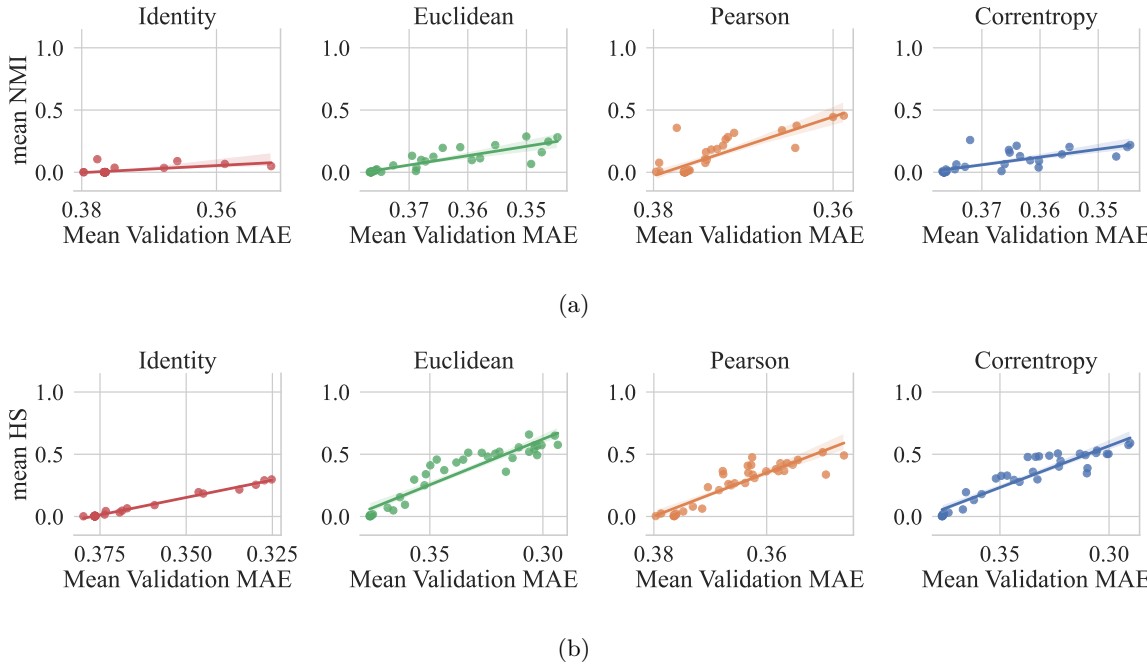

Figure 7: Linear regression plot for CER (the x-axis is inverted) with (a): 2 clusters and (b): 5 clusters.

emphasis on the shape rather than on the magnitude, one should normalize each time series by its own mean and standard deviation.

**Unsupervised model selection** As in the synthetic dataset experiment, in Fig. 7 we display the relationship between NMI/HS and the validation MAE in the settings where the graph is built using the time series similarities (Euclidean, Pearson, and Correntropy). For comparison, we also include the setting where the

Table 3: Ablation results based on hyperparameter search over the loss coefficients. For each configuration, the first three columns contain the NMI, HS, and val MAE obtained by selecting the model setup with the lowest val MAE (based on mean values from 3 runs). The next two columns give the highest observed NMI and HS values in the search. The last two columns present correlation metrics between NMI and val MAE, where $r_p$: Pearson correlation coefficient and $r_s$: Spearman's rank correlation coefficient.

| Config | Obtained from search | | | Best NMI and HS | | NMI-MAE corr. | |
|---|---|---|---|---|---|---|---|
| | NMI | HS | MAE | NMI | HS | $r_p$ | $r_s$ |
| AB1 | $85.2_{\pm2.1}$ | $86.2_{\pm1.9}$ | $0.757_{\pm0.001}$ | $88.2_{\pm1.2}$ | $88.7_{\pm1.1}$ | -0.959 | -0.968 |
| AB2 | $95.1_{\pm0.2}$ | $95.4_{\pm0.1}$ | $0.685_{\pm0.001}$ | $95.4_{\pm0.1}$ | $95.1_{\pm0.2}$ | -0.922 | -0.886 |
| AB3 | $51.0_{\pm3.8}$ | $52.7_{\pm3.9}$ | $0.731_{\pm0.005}$ | $57.0_{\pm28.1}$ | $57.2_{\pm31.2}$ | -0.583 | -0.570 |
| AB4 | $75.2_{\pm7.3}$ | $77.6_{\pm6.8}$ | $0.698_{\pm0.011}$ | $75.2_{\pm7.3}$ | $77.6_{\pm6.8}$ | -0.743 | -0.638 |
| Proposed | $95.5_{\pm0.2}$ | $95.8_{\pm0.2}$ | $0.682_{\pm0.000}$ | $95.5_{\pm0.2}$ | $95.8_{\pm0.2}$ | -0.915 | -0.863 |

graph is represented by the identity matrix, which corresponds to a configuration where no graph information is given. Also in this case, there is a pronounced correlation between the two metrics, which empirically supports our hypothesis of leveraging MAE for doing model selection in the absence of supervision.

The low dispersion of the points around the linear fit indicates that there is a strong relationship between NMI/HS and MAE, no matter how the graph is constructed. However, the numerical values achieved by the different graph construction methods cannot be compared directly. For example, the configuration that achieves the lowest mean MAE does not necessarily imply the best NMI, as evidenced by the comparison between Euclidean and Pearson in Fig. 7a.

## 6.3 Additional ablation studies

The experiments on the synthetic data and CER have shown the effect of different design choices, including testing different loss functions and coefficient values, utilizing different adjacency construction methods, and training with only the forecasting loss. In this section, we report further ablation experiments aimed at studying different designs of the sequence encoder and different types of scheduling for the softmax temperature. The results are limited to the Balanced-NU dataset, which is the more complex of the two balanced datasets. The tested configurations are as follows:

- AB1: A configuration where the TCN in the encoder is removed and a basic mean aggregator is used to reduce the temporal dimension before message passing.

- AB2: A configuration that uses a mean aggregator after TCN instead of an attention-based aggregator.

- AB3 and AB4: Configurations with static softmax temperature during training, fixed at 1.0 and 0.01, respectively.

The results of each ablation configuration are presented in Table 3. The performance of AB1 and AB2 shows that utilizing a deep learning based sequence encoder is beneficial to the clustering, but in this case, there is no significant difference between using a TCN with mean or attention-based aggregation. AB4 and AB5 show the benefit of the temperature scheduling in the softmax used to produce the cluster assignments and subsequently perform pooling. In particular, fixing the temperature at 1 results in significantly worse cluster-class correspondence and a worse NMI-MAE correlation.

## 6.4 Additional configuration experiments

Here, we report additional experiments whose goal is to investigate further the importance and impact of the chosen sequence encoder, MP layers, and temporal decoder by replacing them with different components. In particular, we consider the following variants:

Table 4: Additional experimental results. The experimental setup is the same as for Table 3.

| Config | Obtained from search | | | Best NMI and HS | | NMI-MAE corr. | |
|---|---|---|---|---|---|---|---|
| | NMI | HS | MAE | NMI | HS | $r_p$ | $r_s$ |
| RNN-STGNNEncoder | $95.7_{\pm 0.1}$ | $96.0_{\pm 0.1}$ | $0.683_{\pm 0.000}$ | $95.7_{\pm 0.1}$ | $96.0_{\pm 0.1}$ | -0.979 | -0.993 |
| GAT-STGNNEncoder | $95.1_{\pm 0.1}$ | $95.5_{\pm 0.1}$ | $0.683_{\pm 0.000}$ | $95.3_{\pm 0.1}$ | $95.6_{\pm 0.1}$ | -0.940 | -0.974 |
| EncDecSkip | $72.5_{\pm 0.5}$ | $72.9_{\pm 0.5}$ | $0.563_{\pm 0.001}$ | $73.3_{\pm 0.8}$ | $73.7_{\pm 0.8}$ | -0.838 | -0.759 |
| EmbeddingDecoder | $14.6_{\pm 0.9}$ | $15.2_{\pm 0.9}$ | $0.715_{\pm 0.004}$ | $86.9_{\pm 10.8}$ | $87.2_{\pm 10.7}$ | -0.303 | -0.272 |
| Proposed | $95.5_{\pm 0.2}$ | $95.8_{\pm 0.2}$ | $0.682_{\pm 0.000}$ | $95.5_{\pm 0.2}$ | $95.8_{\pm 0.2}$ | -0.915 | -0.863 |

- RNN-STGNNEncoder: In this variant, the TCN is replaced with a GRU RNN (Cho et al., 2014) with the same hidden size and number of layers, where the contraction of the temporal dimension is accomplished by using only the last state of the RNN. While we have already established that using a deep learning based sequence encoding improves the performance, it is important to check if using RNNs can improve both forecasting and clustering performance. Indeed, RNNs can outperform TCNs in some cases and learn more impactful latent embeddings.

- GAT-STGNNEncoder: This variant replaces the GCS layers in the encoder with GAT layers (Veličković et al., 2018). The purpose of testing this configuration is to ascertain to what extent adopting a more sophisticated attention-based MP approach influences the clustering outcome.

- EncDecSkip: In this variant, we add a skip connection that bypasses the bottleneck between the encoder and decoder. Specifically, in addition to being passed to the pooling operator, the embeddings from the MP layers are passed to a different MLP that reintroduces the temporal dimension. The output of this second decoder is added to the output of the lift operator to form the final prediction. This variant investigates the role and the importance of the bottleneck by incorporating a computational path that bypasses it entirely.

- EmbeddingDecoder: This is a variant that retains the bottleneck without any bypass option, but increases the capacity of the components processing the pooled representations in the decoder. Specifically, the number of temporal channels in the latent factors is increased from 1 to 8, and, after the lift operation, a learned embedding with dimensions $N \times 8$ is used to weight the different channels and produce a final prediction for each node. The rationale behind this architecture is to test whether having different predictions for each time series (obtained by using local node embeddings) would improve or degrade clustering performance.

The experimental setup is otherwise the same as with the additional ablations of Section 6.3, and the results are reported in Table 4. Starting with the variant using an RNN sequence encoder (RNN-STGNNEncoder), we see comparable clustering performance and an improvement in the correlation metrics. This latter improvement indicates a more monotonic relationship between NMI and MAE, and a possible higher guarantee that basing model selection on MAE performance will result in higher cluster quality. However, as we already mentioned, the resulting cluster-class correspondence is not significantly improved, and the RNN-based model requires a 35% longer training time on the same machine with an RTX 4090 GPU. So, for this particular dataset, we argue that the TCN-based model is preferable. Replacing GCS with GAT (GAT-STGNNEncoder) does not significantly change the clustering performance, although the correlation between NMI and MAE becomes stronger, while keeping the computational complexity comparable with GCS.

Regarding EncDecSkip, introducing a skip connection to bypass the bottleneck resulted in improved forecasting performance, which is expected. However, the clustering performance is degraded, and the NMI-MAE correlation is weaker. On the other hand, for EmbeddingDecoder, the added capacity via more channels and additional local embeddings did not enhance the forecasting performance. In addition to the lack of improved predictive performance, this variant produced a significant detrimental effect on the NMI-MAE correlation, resulting in a model selection that yields poor clustering performance. In particular, selecting the model with the lowest MAE also resulted in much lower NMI and HS than the highest observed in the

hyperparameter search, further illustrating that the model selection process becomes unreliable. In both of the latter two variants, a plausible reason for the weakened correlation is that the clustering and forecasting objectives become less aligned when the architecture becomes more suitable for forecasting.

## 7 Conclusions

In this work, we have introduced a framework for clustering time series using spatio-temporal graph neural networks (STGNNs). Our findings highlight the importance of considering not only the temporal features of the individual time series but also the spatial axis describing the relationships across the time series to find meaningful clustering partitions. By employing an encoder-decoder architecture with intermediate and interpretable latent factors, the proposed framework adjusts to the underlying data structures and manages to form meaningful clusters in different scenarios.

Through experimentation with both synthetic and real-world datasets, we have validated the robustness and flexibility of our model. A series of quantitative and qualitative analyses underscores the model's capability to leverage structural and temporal data, thereby forming informative clusters that show a good correspondence with the class labels.

The optimization of the loss coefficients is a vital component in learning a meaningful clustering partition. In particular, several of the existing pooling losses promote a balanced partition, and adjusting the weights of the different components of the loss can lead to the formation of clusters with a non-uniform size. However, finding the optimal hyperparameter configurations without relying on supervised cross-validation is a well-known challenge. To address this issue, we showed that the self-supervised forecasting loss function can act as an efficient tool for guiding the clustering partition. This important connection allows us to perform model selection in a completely unsupervised setting.

Clustering spatio-temporal data using STGNNs remains a largely unexplored area, and our study represents an important initial step toward systematically addressing this gap. Through extensive quantitative and qualitative analyses, we have demonstrated the feasibility and potential of STGNN-based clustering. To continue the research in this direction, there is an urgent need for standardized benchmarking frameworks and comprehensive datasets to more effectively evaluate and compare emerging methodologies and STGNN architectures. While further research and additional real-world datasets are essential, we believe our proposed evaluation framework and the introduced synthetic dataset provide a solid foundation, facilitating future advancements and inspiring continued exploration in this promising research direction.

### Acknowledgments

This work was supported by the Norwegian Research Council grant no. 345017 (*RELAY: Relational Deep Learning for Energy Analytics*). We wish to thank Nvidia Corporation for donating some of the GPUs used in this project.

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

# A    Additional background material

Table 5 summarizes the components of the pooling losses considered in this work. The implementation of DMoN in Pytorch Geometric[1] also uses the orthogonality loss of MinCutPool, but we consider only the modularity and collapse regularization losses that are proposed in the original paper. We refer the interested readers to the reference papers for a more in-depth explanation about the different pooling losses.

Table 5: Pooling loss components.

| Method | Topology-oriented loss | Quality-oriented loss |
|---|---|---|
| MinCutPool (Bianchi et al., 2020a) | $-\dfrac{\mathrm{Tr}(\boldsymbol{S}^\top \boldsymbol{A}\boldsymbol{S})}{\mathrm{Tr}(\boldsymbol{S}^\top \boldsymbol{D}\boldsymbol{S})}$ | $\left\|\dfrac{\boldsymbol{S}^\top \boldsymbol{S}}{\|\boldsymbol{S}^\top \boldsymbol{S}\|_F} - \dfrac{\boldsymbol{I}_C}{\sqrt{C}}\right\|_F$ |
| DiffPool (Ying et al., 2018) | $\left\|\boldsymbol{A} - \mathrm{softmax}(\boldsymbol{S})\mathrm{softmax}(\boldsymbol{S})^\top\right\|_F$ | $\dfrac{1}{N}\sum_{n=1}^{N} H(\boldsymbol{S}_n)\boldsymbol{S}$ |
| DMoN (Tsitsulin et al., 2020) | $-\dfrac{1}{2m}\cdot\mathrm{Tr}(\boldsymbol{S}^\top \boldsymbol{B}\boldsymbol{S})$ | $\dfrac{\sqrt{C}}{n}\left\|\sum_i \mathbf{C}_i^\top\right\|_F - 1$ |
| TVGNN (Hansen & Bianchi, 2023) | $\dfrac{1}{2E}\sum_{i,j,k} A_{i,j}\lvert S_{i,k} - S_{j,k}\rvert$ | $\dfrac{N(K-1) - \sum_k \|\boldsymbol{S}_{:,k} - \mathrm{quant}_{K-1}(\boldsymbol{S}_{:,k})\|_{1,K-1}}{N(K-1)}$ |

# B    Dataset details

In this section, additional details about the applied datasets are given.

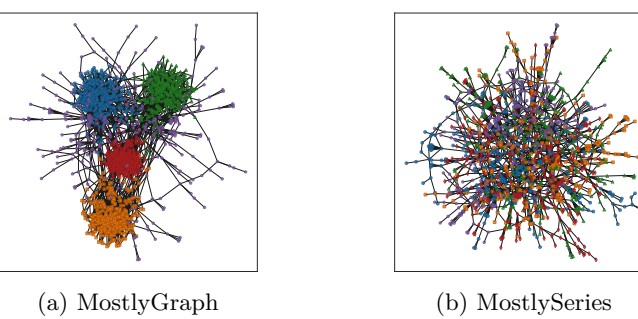

(a) MostlyGraph             (b) MostlySeries

Figure 8: Visual representation of the graphs of MostlyGraph and MostlySeries. The colors/shapes correspond to the different underlying classes. Node positions were generated using the Fruchterman-Reingold force-directed algorithm.

## B.1    Synthetic data

In addition to the four synthetic datasets presented in Section 6.1, two more variants were constructed: *MostlyGraph* and *MostlySeries*. As the names suggest, these two datasets incorporate more class-related information into either the series or the graph compared to the two balanced datasets (Balanced-U and Balanced-NU). The graphs of MostlySeries and MostlyGraph are depicted in Fig. 8.

The parameters utilized to generate the six datasets are presented in Table 6. Each dataset comprises 1500 nodes and 2000 time steps. In the table, $C_{\mathrm{distr}}$ represents the class distribution, $m$ and $m_0$ denote the BA parameters, $\boldsymbol{\Theta}$, $a$, $\phi$, $p$, and $\sigma$ are the series parameters presented in Eqs. 16 and 17. For the OnlyGraph dataset, $\boldsymbol{\theta}$ contains the AR parameters of the shared AR(2) model employed to generate the series. For

---

[1] https://pytorch-geometric.readthedocs.io/en/latest/generated/torch_geometric.nn.dense.DMoNPooling.html

OnlySeries, the parameter `prob` denotes the probability of a connection between two nodes in the Erdős-Rényi graph, and `connected = True` indicates that after all initial edges are sampled, additional edges are randomly added until the graph is connected.

Table 6: Parameters for the synthetic datasets

| **Balanced-U** | **Balanced-NU** | **MostlyGraph** |
|---|---|---|
| $C_{\text{distr}}$: Uniform | $C_{\text{distr}}$ : [.4, .2, .1, .15, .15] | $C_{\text{distr}}$: Uniform |
| $m : [2, 1, 4, 1, 2]$ | Rest the same as Balanced-U | $m : [2, 2, 2, 3, 1]$ |
| $m_0 : [3, 2, 5, 2, 4]$ | | $m_0 : [3, 3, 3, 3, 1]$ |
| $\Theta = \begin{pmatrix} .6 & 0 \\ 0 & .2 \end{pmatrix}$ | | $\Theta$: Same as Balanced-U |
| $a : [.4, .5, .6, .4, .5]$ | | $a : [.5, .5, .5, .5, .5]$ |
| $\phi : [.00, .15, -.15, -.20, .20]$ | | $\phi : [.2, .2, .2, .2, .2]$ |
| $p : [12, 14, 10, 16, 12]$ | | $p : [12, 12, 12, 12, 12]$ |
| $\sigma = .6$ | | $\sigma = .6$ |
| **MostlySeries** | **OnlyGraph** | **OnlySeries** |
| $C_{\text{distr}}$: Uniform | $C_{\text{distr}}$: Uniform | $C_{\text{distr}}$: Uniform |
| $m : [1, 1, 1, 1, 1]$ | $m : [2, 2, 2, 2, 2]$ | $\texttt{prob} = .005$ |
| $m_0 : [2, 2, 2, 2, 2]$ | $m_0 : [3, 3, 3, 3, 3]$ | $\texttt{connected} = \text{True}$ |
| $\Theta = \begin{pmatrix} .6 & .1 \\ .3 & .2 \end{pmatrix}$ | $\theta = (.5, .3)$ | $\Theta$: Same as MostlySeries |
| $a : [.8, .8, .8, .8, .8]$ | N/A | $a : [1.2, 1.5, .8, 2.0, 1.4]$ |
| $\phi : [.0, .15, -.15, -.20, .20]$ | N/A | $\phi : [.0, .15, -.15, -.20, .20]$ |
| $p: [12, 14, 8, 16, 20]$ | N/A | $p : [12, 14, 8, 16, 20]$ |
| $\sigma = .6$ | $\sigma = .6$ | $\sigma = .4$ |

### B.2 Real world load data

The sub-sampled and filtered CER dataset consists of 1,541 nodes, where 1,056 are "residential" and 485 are "SME". The time series of each node consists of 12,865 values after an hourly resampling procedure. The constructed graphs using Euclidean distance, Pearson correlation, correntropy, and Erdős-Rényi are displayed in Fig. 9.

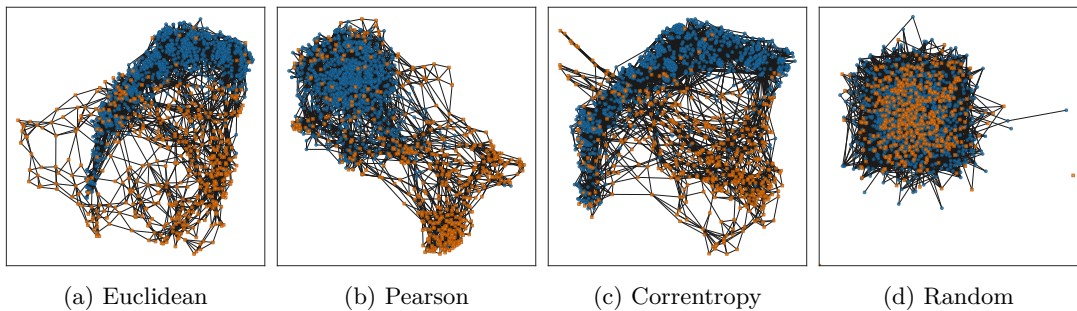

(a) Euclidean      (b) Pearson      (c) Correntropy      (d) Random

Figure 9: Visual representation of the graphs computed for the CER dataset when using four different adjacency construction methods. The colors/shapes correspond to the different underlying classes: blue round nodes denote "residential" and squared orange nodes the "SME" classes, respectively. Node positions were generated using the Fruchterman-Reingold force-directed algorithm.

## C  Experimental details and additional results

This section presents the model and training parameters used in the experiments and some additional results.

Table 7: Pooling loss coefficients for the model with MinCutPool obtained from hyperparameter searches. $c_1$: topology oriented, $c_2$: regularization oriented.

| Synthetic | | | CER | | | | |
|---|---|---|---|---|---|---|---|
| **Dataset** | **5 clusters** | | **Adjacency** | **2 clusters** | | **5 clusters** | |
| | $c_1$ | $c_2$ | | $c_1$ | $c_2$ | $c_1$ | $c_2$ |
| Balanced-U | 0.10 | 0.10 | Identity | 1.06 | 0.10 | 2.02 | 0.10 |
| Balanced-NU | 1.06 | 0.10 | Full | 2.50 | 0.58 | 1.06 | 2.50 |
| MostlySeries | 0.10 | 0.10 | Random | 0.10 | 0.10 | 0.10 | 1.54 |
| MostlyGraph | 0.58 | 0.10 | Euclidean | 2.02 | 0.10 | 1.06 | 0.10 |
| OnlySeries | 0.10 | 0.10 | Pearson | 2.50 | 0.10 | 0.58 | 0.10 |
| OnlyGraph | 1.54 | 0.58 | Correntropy | 2.50 | 0.10 | 2.50 | 0.58 |

## C.1   Model and training parameters

**Model configuration** In each layer of the model, the hidden size $H$ is fixed at 16. The MLP components of the conditional block comprise a single hidden layer and an output layer, both with the same size $H$ and ReLU activation functions. The bias term is omitted from the output layer of the MLP processing the covariates. The TCN used as the temporal encoder consists of two temporal layers, with the dilation of the second layer set to 2. The ReLU function is used as the activation in each layer of the TCN. The hidden and output sizes are set to $H$, and the kernel size of the temporal convolution is set to 3. The TCN is also configured with a skip connection between the temporal layers. Finally, the temporal dimension is aggregated using a linear attention layer. The GNN encoder comprises two message passing layers, implemented as described in Eq. 4, with hidden and output sizes $H$. The temporal decoder is an MLP with a single hidden layer of size $H$ and an output size $T_H = 1$.

**Training procedure** In all experiments and configurations, the model was trained for 250 epochs with Adam using a multi-step learning rate scheduler reducing the learning rate by a factor of 0.5 every 50 epochs, with the initial learning rate set to 0.001. The optimizer was also configured with weight decay regularization (L2 penalty) set to 0.0001.

The softmax temperature used when computing the cluster assignment matrix was initially set to 1.0 and then decreased linearly by 0.0099 per epoch down to a minimum of 0.01.

Gradient clipping was applied during backpropagation with a clip value of 5.

**Hyperparameter search** In all experiments, a grid search was conducted to find suitable values for the pooling loss coefficients where $c_1, c_2 \in \{0.1, 0.58, 1.06, 1.54, 2.02, 2.5\}$ (six linearly spaced points between 0.1 and 2.5). The resulting coefficients from the searches are reported in Table 7

## C.2   Synthetic data experiment

For the synthetic datasets, the proposed model was trained with a batch size of 16. The model was not given covariates, so the CondBlock of Eq. 9 was comprised of a single MLP layer with an output size of 16 and ReLU activation.

The obtained numeric results for MostlyGraph and MostlySeries are presented in Table 8. The results are aligned with those presented in Section 6.1 for the other dataset configurations, where the proposed model better handles the combination of spatial and temporal features, even compared to the methods applied to the graph with extracted features, as they rely more heavily on the presence of relevant features in both the graph structure and node attributes to produce meaningful clusters.

The results obtained for the different pooling losses are reported in Table 9. With the exception of Diff-Pool, the different losses produce similar results. TVGNN differs significantly from the top performers for OnlyGraph. However, for this dataset, the time series do, by intent, not contain any distinctive features to help guide the clustering. All distinctive features are in the graph, and therefore, the correlation between forecasting performance and cluster-class correspondence does not hold, as can be seen in Fig. 10. The model

Table 8: Additional synthetic data experiment results.

| Method | | MostlySeries | | | MostlyGraph | | |
|---|---|---|---|---|---|---|---|
| | | NMI | HS | CS | NMI | HS | CS |
| **Spatial** | SC | $16.9_{\pm0.0}$ | $13.6_{\pm0.0}$ | $22.4_{\pm0.0}$ | $81.5_{\pm0.0}$ | $80.5_{\pm0.0}$ | $82.6_{\pm0.0}$ |
| | Walktrap | $34.6_{\pm0.0}$ | $65.1_{\pm0.0}$ | $23.6_{\pm0.0}$ | $74.6_{\pm0.0}$ | $\underline{90.9}_{\pm0.0}$ | $63.2_{\pm0.0}$ |
| | Leiden | $32.2_{\pm0.0}$ | $52.3_{\pm0.0}$ | $23.2_{\pm0.0}$ | $\mathbf{90.8}_{\pm0.0}$ | $\mathbf{99.5}_{\pm0.0}$ | $\underline{83.5}_{\pm0.0}$ |
| **Temporal** | Euclidean | $\mathbf{100.0}_{\pm0.0}$ | $\mathbf{100.0}_{\pm0.0}$ | $\mathbf{100.0}_{\pm0.0}$ | $43.5_{\pm0.0}$ | $42.5_{\pm0.0}$ | $44.7_{\pm0.0}$ |
| | Cosine | $\mathbf{100.0}_{\pm0.0}$ | $\mathbf{100.0}_{\pm0.0}$ | $\mathbf{100.0}_{\pm0.0}$ | $49.1_{\pm0.1}$ | $49.0_{\pm0.1}$ | $49.2_{\pm0.1}$ |
| | DTW | $74.2_{\pm0.0}$ | $71.4_{\pm0.0}$ | $77.2_{\pm0.0}$ | $23.7_{\pm0.1}$ | $22.6_{\pm0.1}$ | $25.1_{\pm0.1}$ |
| | ESN | $\underline{99.8}_{\pm0.2}$ | $\underline{99.8}_{\pm0.2}$ | $\underline{99.8}_{\pm0.2}$ | $6.2_{\pm1.4}$ | $5.8_{\pm1.3}$ | $6.8_{\pm1.5}$ |
| **Static feats** | GAE | $95.2_{\pm0.9}$ | $95.2_{\pm0.9}$ | $95.2_{\pm0.9}$ | $39.8_{\pm3.8}$ | $34.8_{\pm3.8}$ | $46.6_{\pm4.2}$ |
| | VGAE | $92.3_{\pm1.4}$ | $92.2_{\pm1.4}$ | $92.3_{\pm1.4}$ | $47.0_{\pm3.8}$ | $44.7_{\pm5.4}$ | $49.8_{\pm2.5}$ |
| | StaticPool | $84.6_{\pm4.7}$ | $84.3_{\pm5.0}$ | $85.0_{\pm4.4}$ | $39.7_{\pm2.2}$ | $39.6_{\pm2.1}$ | $39.8_{\pm2.3}$ |
| **Proposed** | NoPoolLoss | $\mathbf{100.0}_{\pm0.0}$ | $\mathbf{100.0}_{\pm0.0}$ | $\mathbf{100.0}_{\pm0.0}$ | $46.4_{\pm2.8}$ | $45.5_{\pm2.7}$ | $47.5_{\pm2.9}$ |
| | MinCutPool | $\mathbf{100.0}_{\pm0.0}$ | $\mathbf{100.0}_{\pm0.0}$ | $\mathbf{100.0}_{\pm0.0}$ | $\underline{86.8}_{\pm6.7}$ | $86.8_{\pm6.7}$ | $\mathbf{86.9}_{\pm6.7}$ |

Table 9: Synthetic data results for different pooling methods.

| Method | Balanced-U | | | Balanced-NU | | | OnlySeries | | |
|---|---|---|---|---|---|---|---|---|---|
| | NMI | HS | CS | NMI | HS | CS | NMI | HS | CS |
| NoPoolLoss | $74.2_{\pm37.1}$ | $73.9_{\pm37.0}$ | $94.6_{\pm2.9}$ | $83.6_{\pm1.3}$ | $85.0_{\pm1.5}$ | $82.2_{\pm1.2}$ | $\mathbf{100.0}_{\pm0.0}$ | $\mathbf{100.0}_{\pm0.0}$ | $\mathbf{100.0}_{\pm0.0}$ |
| DiffPool | $64.0_{\pm32.2}$ | $59.8_{\pm30.7}$ | $69.2_{\pm34.4}$ | $58.6_{\pm29.2}$ | $55.7_{\pm28.0}$ | $61.7_{\pm30.9}$ | $\underline{97.3}_{\pm5.4}$ | $\underline{96.6}_{\pm6.9}$ | $\underline{98.1}_{\pm3.8}$ |
| MinCutPool | $\mathbf{100.0}_{\pm0.0}$ | $\mathbf{100.0}_{\pm0.0}$ | $\mathbf{100.0}_{\pm0.0}$ | $\underline{95.5}_{\pm0.2}$ | $\underline{95.8}_{\pm0.2}$ | $\underline{95.3}_{\pm0.2}$ | $75.1_{\pm38.6}$ | $74.7_{\pm38.6}$ | $75.5_{\pm38.6}$ |
| DMoN | $\underline{96.5}_{\pm6.8}$ | $\underline{96.5}_{\pm6.8}$ | $\underline{96.5}_{\pm6.8}$ | $86.7_{\pm1.4}$ | $88.3_{\pm1.4}$ | $85.1_{\pm1.5}$ | $\mathbf{100.0}_{\pm0.0}$ | $\mathbf{100.0}_{\pm0.0}$ | $\mathbf{100.0}_{\pm0.0}$ |
| TVGNN | $93.3_{\pm8.2}$ | $93.3_{\pm8.2}$ | $93.3_{\pm8.2}$ | $\mathbf{99.9}_{\pm0.1}$ | $\mathbf{99.9}_{\pm0.1}$ | $\mathbf{99.1}_{\pm0.1}$ | $\mathbf{100.0}_{\pm0.0}$ | $\mathbf{100.0}_{\pm0.0}$ | $\mathbf{100.0}_{\pm0.0}$ |
| | OnlyGraph | | | MostlySeries | | | MostlyGraph | | |
| | NMI | HS | CS | NMI | HS | CS | NMI | HS | CS |
| NoPoolLoss | $0.0_{\pm0.0}$ | $0.0_{\pm0.0}$ | $0.0_{\pm0.0}$ | $\mathbf{100.0}_{\pm0.0}$ | $\mathbf{100.0}_{\pm0.0}$ | $\mathbf{100.0}_{\pm0.0}$ | $46.4_{\pm2.8}$ | $45.5_{\pm2.7}$ | $47.5_{\pm2.9}$ |
| DiffPool | $0.4_{\pm0.1}$ | $0.4_{\pm0.1}$ | $0.4_{\pm0.1}$ | $\underline{93.2}_{\pm5.7}$ | $\underline{89.7}_{\pm8.4}$ | $\underline{97.4}_{\pm3.5}$ | $17.0_{\pm3.8}$ | $16.8_{\pm3.8}$ | $17.2_{\pm3.8}$ |
| MinCutPool | $\mathbf{100.0}_{\pm0.0}$ | $\mathbf{100.0}_{\pm0.0}$ | $\mathbf{100.0}_{\pm0.0}$ | $\mathbf{100.0}_{\pm0.0}$ | $\mathbf{100.0}_{\pm0.0}$ | $\mathbf{100.0}_{\pm0.0}$ | $86.8_{\pm6.7}$ | $86.8_{\pm6.7}$ | $86.9_{\pm6.7}$ |
| DMoN | $\mathbf{100.0}_{\pm0.0}$ | $\mathbf{100.0}_{\pm0.0}$ | $\mathbf{100.0}_{\pm0.0}$ | $\mathbf{100.0}_{\pm0.0}$ | $\mathbf{100.0}_{\pm0.0}$ | $\mathbf{100.0}_{\pm0.0}$ | $\underline{93.9}_{\pm10.8}$ | $\underline{93.9}_{\pm10.8}$ | $\underline{93.9}_{\pm10.8}$ |
| TVGNN | $\underline{1.3}_{\pm0.3}$ | $\underline{1.3}_{\pm0.3}$ | $\underline{1.3}_{\pm0.3}$ | $\mathbf{100.0}_{\pm0.0}$ | $\mathbf{100.0}_{\pm0.0}$ | $\mathbf{100.0}_{\pm0.0}$ | $\mathbf{98.5}_{\pm2.4}$ | $\mathbf{98.5}_{\pm2.4}$ | $\mathbf{98.5}_{\pm2.3}$ |

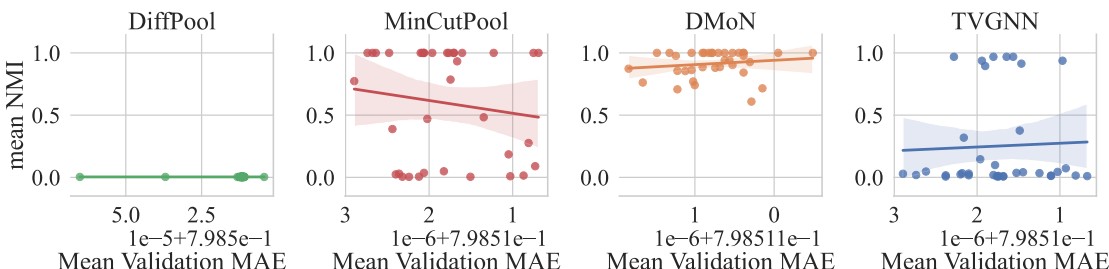

Figure 10: NMI-MAE correlation for OnlyGraph.

selection based on forecasting MAE is thus ineffective, as expected. It is also noteworthy that the different losses exhibit significantly different behaviors within the selected search space for the loss coefficients, with DMoN consistently delivering less erratic outcomes than the other two graph-cut related pooling losses, and DiffPool consistently underperforms.

### C.3 Real world data experiment

There are a few differences in the training setup and model configuration used in the real-world data experiment. Firstly, a set of exogenous variables, $\boldsymbol{U}_{t:t+T}$, based on sinusoidal functions with daily, weekly, and yearly periods was utilized to help the model better capture the seasonal patterns in the data. Secondly, the batch size was set to 8.

Lastly, in the lift operation of the model, the softmax with temperature is replaced with a hardmax using the straight-through trick. More specifically, during forward propagation, the hardmax is applied, while the softmax with temperature is applied during backpropagation. This change was made due to the presence of multiple outliers, which produced a low validation MAE when a near-hard cluster assignment matrix, which had also collapsed into a single cluster, was combined with high-magnitude latent factors corresponding to the near-zero assignment values. This phenomenon is illustrated in Fig. 11.

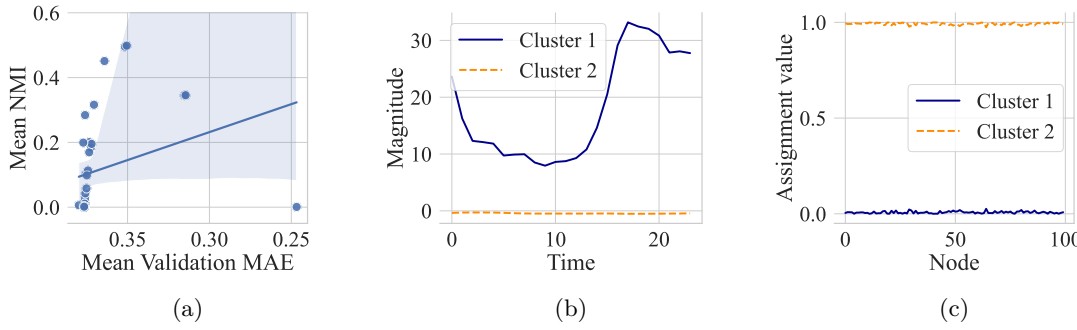

(a)   (b)   (c)

Figure 11: The effect of not using the straight-through trick on CER with Pearson adjacency. (a): NMI-MAE correlation plot, (b): Latent factor snippet of each cluster for the rightmost outlier in (a), (c): Soft assignments for the first 100 nodes for the same outlier.

