# OpenReview forum: "On Time Series Clustering with Graph Neural Networks"
_TMLR — Accepted by TMLR_

### Review · Reviewer_jB4K · 2025-06-06

**Summary Of Contributions:**

- The paper demonstrates how existing graph pooling methods (MinCutPool, DiffPool, DMoN, TVGNN) can be incorporated into a spatio-temporal GNN architecture to cluster multivariate time series.
The authors show that spatial graph structure can improve clustering performance even when artificially constructed from temporal similarities.

- The authors empirically show that forecasting loss correlates with clustering quality metrics in their experimental settings.

- The paper introduces a set of synthetic datasets where class membership is determined by varying combinations of graph topology and temporal dynamics. The authors provide controlled test cases for evaluating spatio-temporal clustering methods.

**Audience:**

No

**Claims And Evidence:**

Yes

**Requested Changes:**

- Q1. The authors mention on page 7 that "during training, we gradually lower the temperature $\tau$ in the softmax used to compute the soft cluster assignments $S$," which suggests the use of Gumbel softmax in Eq.(12), similar to Cini et al. (2024). However, the authors write as if they are not using Gumbel softmax. Could you clarify whether Gumbel softmax is actually being used?

- Q2. The authors claim in the introduction that "the forecasting accuracy correlates well with the quality of the extracted clusters." However, Figure 10 shows that this correlation does not appear for  OnlyGraph dataset. This requires further analysis, or the claim in the introduction should be revised to reflect this limitation.

- Q3. The claim that prediction loss correlates with clustering quality for unsupervised model selection relies solely on empirical evidence. Can the authors provide theoretical justification or cite related work that supports why this correlation should hold in general?

- Q4. Can the authors provide ablation studies on their architectural choices to justify key design decisions?

---

- Minor comment: The explanation of $\eta_t$ in Eq.(17) is not sufficiently clear and should be expanded for better readability.

**Strengths And Weaknesses:**

**Strengths**

- The paper provides empirical evidence that forecasting MAE can serve as a proxy for clustering quality and offers a practical solution for hyperparameter tuning without ground truth labels.

-  The synthetic datasets vary the importance of spatial vs temporal information, thereby enabling controlled evaluation of when graph structure helps clustering performance.

**Weaknesses**

- The actual contribution amounts to applying existing graph pooling methods to time series data. The paper provides no new insights into *how* or *why* these methods are effective for spatio-temporal clustering.

- Only one real-world dataset (CER smart meter) is used, and crucially, this dataset has no natural spatial relationships. The authors artificially construct a graph using similarity measures, which contradicts their claim in the abstract: *"can leverage prior knowledge on the spatial structure of the data"*

- Section 3.1 fails to provide clear reasoning or motivation for why the proposed model uses pooling operators in this specific configuration.

- The authors provide no ablation studies on their architectural choices, leaving key design decisions unjustified.

---

> ### Author Response · Authors · 2025-07-02
>
> Thank you for your review and for the useful comments and suggestions. This is much appreciated!
>
> Below, we reply point-by-point to each weakness and requested change.
>
> ### Weaknesses
>
> **W1:** Graph pooling is an important part of our architecture. However, our actual contribution is on how to perform clustering of spatio-temporal data with a deep learning architecture, rather than testing existing pooling operators in a different setting. The proposed clustering framework should be considered as a whole: every component has been carefully chosen to achieve our goals, while keeping simplicity in mind. We would like to stress that the graph pooling operators are basically the only option for implementing this kind of GNN framework. As such, the focus should be more on the how and why the whole framework works, rather than on the pooling operators that are just a part of it. Answering those questions is what we want to accomplish with our analyses and experiments.
>
> **W2:** While it is true that CER does not come with any form of intrinsic graph or spatial relationships, the term *prior knowledge* is intended to refer to the inductive biases that can be utilized by leveraging a graph structure. The latter can stem from similarity measures with respect to quantities such as the time series. This is the case for CER, where the prior knowledge is the inductive bias given the graph built from the time series using different similarity measures. We will clarify this in the revised manuscript.
>
> **W3:** There is a strong connection between clustering the nodes of a graph (i.e., spectral clustering) and the family of graph pooling operators that we consider in this work. In particular, we consider those graph pooling operators that reduce the graph by learning cluster assignments that are used to generate supernodes of the pooled graph. Since we consider clustering time series by relying on a spatial component encoded as a graph, these pooling operators become the most natural choice. We will explain this better in the revised manuscript.
>
> **W4:** We respectfully disagree that our work does not contain any ablation study since several of our experiments include configurations that serve as ablation experiments. For instance:
> - the configurations with NoPoolLoss measure the importance of including clustering/pooling losses
> - the identity adjacency in the experiment with CER serves as a configuration without the use of encoded spatial information and message passing
> - we have presented plots and results showing the impact of different pooling losses, and the correlation plots show the effect of varying the pooling loss coefficients.
>
> Nevertheless, we have started work on more ablations to further justify the design decisions. We also welcome requests from the reviewer regarding further ablation experiments for any specific design choices that may appear unjustified.
>
> ### Requested changes
>
> **Q1:** To clarify, we do not use Gumbel softmax since, unlike in the mentioned work by Cini et al., we do not perform discrete sampling. We use softmax solely to produce cluster assignments that should be non-stochastic, so our softmax implementation is the vanilla version with a temperature parameter to control the sharpness of the activation.
>
> **Q2:** There might be a misunderstanding here. OnlyGraph was intentionally designed without distinctive features in the time series, so the lack of correlation is indeed expected here, as our assumptions are built on the presence of such features. We will expand the text referring to Figure 10 to make this point clearer.
>
> **Q3:** Thank you for raising this interesting point. Since there is no universal optimum criterion for clustering performance, the quality of a clustering partition largely depends on prior assumptions and subjective choices. In our case, a key assumption is that higher cluster quality is tied to grouping time series with similar patterns (i.e., that look the same), and design decisions like utilizing forecasting loss as guidance build on this assumption.
>
> Also, in the case of CER, the numeric results obtained with metrics such as NMI build on an assumption that the agreement between cluster and class labels is indicative of a “good” partition, which is another subjective choice. We will revise the text to stress these points.
>
> **Q4:** See W4.
>
> ---
>
> Regarding the minor comment, we agree that the description of  $\eta_t$ is somewhat brief and will expand this for clarity. Thank you for pointing that out.
>
> Finally, we noticed the **Audience: no** at the end of the review, meaning that our work is regarded as not suitable for the TMLR audience. We would like to know if there are some concerns in this respect and if there is anything we could do to address them.

---

> ### Comment · Reviewer_jB4K · 2025-07-03
>
> Thank you for your response. While some of my concerns have been addressed somewhat, I still have additional comments.
>
> Q1. Regarding the authors' response to W1, it seems to conflict with the results of the NoPoolLoss variant. Can you explain why the performance of NoPoolLoss does not worsen? Doesn't this raise questions about the necessity of graph pooling?
>
> Q2. Regarding the response to W2, I still do not understand how you claim to use "prior knowledge of spatial structure" when there is actually no "spatial structure" in the CER dataset. Are the authors using prior knowledge and inductive bias as the same concept, even though they are different concepts? Could you explain this more clearly? Are the authors trying to say that the graphs they create from the temporal similarity of time series are “spatial prior knowledge”?
>
> Q3. Although the authors claim that sufficient ablation studies have already been conducted, there is a lack of comparisons regarding the presence or absence of TCN in the sequence encoder or the use of other temporal encoders, analysis of the contribution of attention-based aggregators, analysis of the impact of temperature scheduling, and systematic analysis of various K (number of clusters) values.

---

> > ### Comment · Reviewer_jB4K · 2025-07-03
> >
> > The "Audience: no" reflects my view that the current version may not fully meet TMLR's standards yet. However, I am open to changing this to "yes" if my remaining concerns are adequately addressed. I want to emphasize that this check is not final and can be revised based on your responses.

---

> > > ### Author Response · Authors · 2025-07-12
> > >
> > > Thank you for your follow-up comments. We appreciate your continued engagement and have addressed your additional points below.
> > >
> > > **To Q1**: To clarify, for NoPoolLoss, the coefficients of the pooling losses are set to zero, meaning that the model is only trained with the forecasting loss, but pooling of the node features is still applied (otherwise we would not obtain any clusters). Furthermore, the NoPoolLoss performs worse than alternatives with graph-based regularizations in all cases except for OnlySeries because the graph in that dataset is completely uninformative.
> > >
> > > **To Q2**: With the term “spatial”, we simply refer to the dimension that spans a collection of time series. With “prior on spatial structure,” we refer to prior beliefs about the (sparse) structure that describes the dependencies among the time series. In the specific case of the CER dataset, our prior translates into the fact that only the observations coming from time series that are correlated with the target should be taken into account when forming clusters and making forecasts. We do agree that our use of the terms "prior information" and "spatial structure" is unclear and requires to clarification, and have added this to the revision (see the revised text in Sections 2.1 and 5.2).
> > >
> > > **To Q3**: We did not mean to suggest that what had been done ablation-wise was necessarily sufficient, we just wanted to clarify that the work already contained some ablation-like testing. We greatly appreciated your feedback regarding additional ablation experiments. In the revised manuscript, we have extended the ablation study based on your suggestions, plus added a study of a few other variants of our architecture (see Sections 6.3 and 6.4 in the revision).

---

> > > > ### Comment · Reviewer_jB4K · 2025-07-14
> > > >
> > > > Thank you for your response. I have read your revised paper and acknowledge that you have addressed my concerns through improved terminology clarification, ablation studies, and expanded experimental comparisons.
> > > > I appreciate the effort put into this revision and will reflect these improvements in my final recommendation.

---

### Review · Reviewer_VMxE · 2025-06-23

**Summary Of Contributions:**

This work investigates the effectiveness of STGNNs for time series clustering, discussing the usefulness of spatial relationships and evaluating STGNNs on tasks with and without an explicit graph structure. This work also evaluates the roles and effects of forecasting and topological loss functions commonly used in temporal graph learning.

**Audience:**

Yes

**Claims And Evidence:**

Yes

**Requested Changes:**

See weaknesses.

**Strengths And Weaknesses:**

**Strengths**

- The paper is well-written and easy to follow.
- This work is largely diagnostic and asks questions about capabilities of the existing STGNN paradigm, which is a valuable contribution besides designing new SOTA models.
- This work conducts comprehensive studies and experiments that generally support the author's conclusions, especially the connection between forecasting MAE and NMI.
- Code is provided.

**Weaknesses**

- This work is mostly focused on a specific proposed STGNN architecture. This makes the conclusions of this work largely specific to their model, and the results would be considerably stronger if other STGNN models were evaluated.
- The central claims of this work are relatively weak. For example, it is well-expected that better forecasting performance should lead to better clustering performance. It is also expected that spatial/temporal information should be heavily exploited when both are provided per the methods' OnlySeries and OnlyGraph performance.
- The real-world dataset validation is rather weak with the max NMI with 2 clusters being less than 50. This is also on a relatively small-size dataset.
- The choice to use sinusoids to model local effects is unjustified. Results would be more robust if other types of noise were benchmarked.

---

> ### Author Response · Authors · 2025-07-02
>
> We sincerely thank you for your effort in reviewing our paper. Your comments and suggestions are very much welcome, and in the following, we did our best to answer your concerns. (Response is split into two comments)
>
> ### Weaknesses
>
> **W1:** While we agree that trying more architectures would allow us to draw stronger conclusions, we would like to stress that, to the best of our knowledge, there are no other STGNN architectures specifically designed to perform time series clustering. There are a couple of recent STGNNs that implement graph pooling internally, but they are designed to perform forecasting, and using them for clustering is not suitable. The reason is that their graph coarsening is either pre-computed [1] (which, then, is equivalent to do spectral clustering on the adjacency matrix), or they have multi-layered architectures with skip connections [2] that forces each latent graph representation to capture details functional to the forecasting rather than the being a proper clustering partition.
>
> However, despite the lack of existing STGNN architectures for clustering, we will aim to add at least one method in our comparison that can utilize information from both data domains by clustering graphs with attributed nodes where the node attributes consist of (static) features extracted from the time series. Nevertheless, if you are aware of any STGNN model suitable for clustering that we missed in our comparison, we are willing to include it in our study.
>
> **W2:** Indeed, we conjectured the presence of a relationship between the forecasting loss and the cluster quality. However, having it manifest in practice and using such a relationship to guide the hyperparameter search was far from trivial, and it took us a significant amount of work.
>
> When we started this work, we considered more complex and larger model configurations, and, in those cases, we were not able to observe any correlation between forecasting accuracy and clustering metrics. The reason was that, thanks to the higher capacity, the models were able to find shortcuts to optimize the forecasting performance without learning meaningful latent factors and clusters. One notable example was when every cluster had basically the same latent factor, and the decoder was the one doing all the heavy lifting. Things started to work when we did several changes, including scaling the decoder complexity down to a minimum and introducing a very narrow bottleneck to encourage learning meaningful clusters with latent representations that serve as good average predictors for forecasting.
>
> So, in summary, while we agree that there is an intuition that motivated the search for such a correlation, it was a difficult endeavor to have it manifest in practice. We believe that, through this paper, we can share our findings with the scientific community and avoid future researchers repeating such efforts. Indeed, we are not aware of previous work that explores such a relationship or that provides more formal results to support it.
>
> Regarding the sentence “It is also expected that spatial/temporal information should be heavily exploited when both are provided per the methods' OnlySeries and OnlyGraph performance” we are not sure we understand your point. Since we care about your opinion and we would like to provide the best possible answer, we kindly ask you to rephrase the comment.
>
> ---
>
> [1] Marisca et al., “Graph-based forecasting with missing data through spatiotemporal downsampling”, ICML, 2024.
>
> [2] Cini et al., “Graph-based Time Series Clustering for End-to-End Hierarchical Forecasting”, ICML, 2024.

---

> > ### Author Response · Authors · 2025-07-02
> >
> > **W3:** First of all, we would like to stress that the series in this dataset are quite long (12865 time points after hourly resampling) and there is some overlap between the two classes in terms of time series attribute distribution, which makes this dataset rather challenging. For example, see Figure 6 in the paper, where for cluster 5, the mean residential and SME profiles are quite similar. Regarding the size in terms of number of nodes, in an unsupervised clustering task such as this, the dataset size matters less than in a supervised classification setting, assuming that we compare differently sized datasets with similar attribute distributions (both in terms of spatial and temporal attributes).
> >
> > Regarding your concern about the performance, we suspect there might be a slight misunderstanding. The NMI is quite sensitive to disagreements between the cluster assignments and the classes. The fact that the class distribution is heavily skewed towards residential also makes the problem more difficult and it is enough to have a few mismatches with respect to the classes to significantly reduce the NMI. If we instead utilize the Kuhn-Munkres algorithm to match up the clusters and classes and compute standard accuracy, the accuracy for the best performer (MinCutPool + Pearson) with 2 clusters averages to about 88.9%, with 93.6% correct assignments for the residential class and 78.5% for SME on average. So, to summarize, an NMI score of ~0.5 is, in this case, **not** indicative of low performance.
> >
> > Finally, we would like to stress that looking at the NMI is, after all, a subjective choice. It is not necessarily true that cluster and class labels should be aligned. For this reason, in the paper, we also put emphasis on qualitative aspects of the clustering and the latent factors to provide a more objective and comprehensive analysis.
> >
> > **W4:** Thank you for the suggestion, we will justify this choice better in the manuscript. Sinusoids represent a basis and, thus, are good choices to be used for expressing more complex signals. Those emerge by combining the signals of neighboring nodes according to the graph topology through the autoregressive component.
> >
> > The use of sinusoids and the graph topology to produce local effects and the use of Gaussian noise to influence intra-class variation is suitable for our purpose of testing the capability of our approach in leveraging both the spatial and temporal components, and making a comparison with methods that only have access to one data component. We do not think that other kinds of noise and local effects could improve the experiment in this sense, nor draw further insights, but we remain open to suggestions.

---

> > > ### Comment · Reviewer_VMxE · 2025-07-07
> > >
> > > Thank you to the authors for their response.
> > >
> > > **W1**: Thank you for aiming to add another method for comparison. While there may be a lack of existing STGNNs for clustering specifically, the general method of learning a cluster assignment using learned node embeddings and decoding them with an MLP seems easily generalizable to temporal GNNs. If the authors can further describe why there aren't naive ways to generalize existing STGNNs for time series clustering, I would be more convinced. I would also appreciate more detail as to how Cini et al.'s architecture is not amenable to learning a proper clustering partition.
> > >
> > > **W2**: Thank you for detailing some of the work building up to this paper. For certain, I do not want to detract from the central claims of this work and the efforts of the authors. I think these details would be very beneficial for the STGNN community, and it would be good if the authors could include some of these details in the appendix. It also seems to suggest that the work's findings should be for low-parameter decoder models specifically, and this should be important for practitioners and other researchers delving into this research area.
> > >
> > > My comment is only noting that spatial/temporal information is used exactly when they should be, when only the time series or only the graph are provided, which is exactly what one would expect to observe in this study. The point of Weakness 2 is that these conclusions are rather weak/unsurprising. I will note though that, as far as I can tell, correctness is not hurt, so this is a minor weakness.
> > >
> > > **W3**: Thank you for clarifying the details of the dataset and the significance of the NMI. This weakness is resolved for me.
> > >
> > > **W4**: Thank you for clarifying your motivation for sinusoidal noise. While I’m not a domain expert in all types of local effects, I’m aware that models such as local linear trend models and other state space approaches are commonly used. My main suggestion is that the work would be more compelling if it considered a broader range of time series types.

---

> > > > ### Author Response · Authors · 2025-07-12
> > > >
> > > > We thank you for your further feedback. Your continued input is appreciated, and we respond to each of the new comments point-by-point below.
> > > >
> > > > **To W1**:  Yes, in principle, we could use any STGNN layer as the encoder, but indeed, this can be seen as different implementations of the same framework and variants of the proposed template architecture. For what concerns the architecture in Cini et al., it is designed with forecasting in mind and has components tailored to that objective, which would complicate a direct comparison with other baselines. Nonetheless, we agree that it is worth exploring additional possible designs, and have therefore done additional experimentation on four variants of our architecture (see Section 6.4 in the revised text).
> > > >
> > > > **To W2**: You are correct in that our findings suggest that the STGNN clustering model seems to benefit from using small and simple decoders. To support this finding with further empirical evidence, we have added two configurations with a higher capacity decoder in the revision (see Section 6.4 in the revision). In short, the results of these configurations show a marked worsening in the correlation between NMI and MAE, and in one case, it is to such a degree that the model selection based on the lowest MAE becomes completely unreliable with respect to clustering. Regarding the remarks about performance on temporal/spatial only datasets, we agree that the results align with what one would expect. Nevertheless, we deemed it important to confirm experimentally that the models behave as they should, in addition to providing quantitative results for a variety of cases, including extreme ones.
> > > >
> > > > **To W4**: Thanks for your suggestion. We would like to clarify that by “local effects” we refer to the local dynamics that might characterize different related time series, as opposed to trends/seasonalities/etc.  In this context, we argue that considering different trends is not particularly informative. Furthermore, trends introduce nonstationarities that often should be dealt with in a separate processing step.

---

> > > > > ### Comment · Reviewer_VMxE · 2025-07-22
> > > > >
> > > > > Thank you to the authors for their revisions.
> > > > >
> > > > > **W1**: Thank you for the additional experiments; these are the results I was looking for. I agree with the analysis and observations detailed in the updated manuscript. This weakness is resolved for me.
> > > > >
> > > > > **W2**: Thank you for these experiments with higher-capacity decoders. My response is similar to weakness 1, and this weakness is resolved for me.
> > > > >
> > > > > **W3**: I think some more argument is needed for why considering different trends is not so informative or relevant for this kind of work. It seems time series data in general can be sensitive to phenomena like trends and seasonality. Some comments in the manuscript would resolve this for me as the main point of the paper is not seriously harmed by this in my view.

---

### Review · Reviewer_gKd4 · 2025-06-29

**Summary Of Contributions:**

This paper proposes a spatiotemporal graph neural network (ST-GNN) architecture and training method for clustering time series in an unsupervised fashion.  That is, the method produces clusters without respect to a downstream task.  The authors show that metrics of clustering quality on their produced clusters correlate well with performance of models trained on these clusters on downstream forecasting tasks.  The authors compare clustering metrics (NMI, HS, CS) of their method with those of various other spatial and temporal time series clustering methods in the literature.

**Audience:**

Yes

**Claims And Evidence:**

No

**Requested Changes:**

Both weaknesses 1 and 2 should be addressed in some fashion.  I feel that both are important for a comprehensive evaluation.  While the results are promising, I do not quite feel that they are comprehensive enough.

**Strengths And Weaknesses:**

Strengths:

1.) The fact that time series can be clustered using both spatial and temporal information in a way that is useful for downstream processing is useful when access to labeled data is limited.

2.) The proposed architecture is a natural generalization of GNNs to spatiotemporal data.

Weaknesses:

1.) Regarding comparison with other methods, it seems odd to compare only with methods that use either spatial or temporal information, but not both.  It seems intuitively obvious to me that having access to both types of data can improve clustering quality.  The performed analysis leaves open the possibility that higher-quality clusters could be produced with a simpler architecture using hand-engineered time series features.

2.)  I do not see much discussion of the reasoning behind the choice of synthetic dataset.  For example, why is the Barabasi-Albert model a good one for the structure of each cluster?  Would the empirical results be the same if, for example, each cluster were an Erdos-Renyi graph with sufficiently large edge probability compared to inter-class probabilities (so the entire graph would be a stochastic block model)?   Potentially, competing methods might perform better if one changes the intra-class graph structure distribution.

---

> ### Author Response · Authors · 2025-07-02
>
> We thank you for your review and for the constructive comments and suggestions. We appreciate your input and provide point-by-point responses to each of the raised concerns below.
>
> ### Weaknesses
>
> **W1:** The intent with the experiments is that access to both data types (spatial and temporal) can improve clustering quality.  We agree that comparing with other methods utilizing both data types would strengthen our conclusions, but we are not aware of any methods designed for this with a focus on clustering. There are other STGNN architectures that use graph pooling/coarsening as part of their framework (see examples under W1 in our reply to reviewer VMxE), but they are typically focused on forecasting and are not suitable for a comparison in clustering performance. However, we think your suggestion of utilizing a simpler architecture with hand-engineered time series features is interesting. We, therefore, aim to add at least one method in our comparison that can cluster nodes in attributed graphs, where the node attributes will consist of static features extracted from the corresponding time series.
>
> **W2:** We have used the Barabasi-Albert (BA) model as opposed to something like the stochastic block model (SBM) due to some properties of the BA model that made it easier to construct graphs with more variation in the edge distribution between the classes/communities. With BA, some of the class subgraphs can appear more tree-like while others can be denser with clear hubs, i.e. select nodes with many edges. The edge sampling done in an SBM, however, mostly just offers different edge densities in the different communities, while other distribution characteristics are largely similar. It is also far easier to ensure a connected graph with BA while ensuring a reasonably sparse graph, while with SBM based graphs, there is a risk of extensive edge sampling before the graph becomes connected (no disjoint graph sections), potentially resulting in a much denser graph.
>
> Regarding the performance of the purely spatial methods, they would likely all perform well on a typical SBM graph with high intra-class edge densities (as long as there are no disconnected components, which they would not be able to handle very well). We would expect performance to be in line with OnlyGraph, and we would also expect our framework to perform similarly well, and it would have an advantage if there are also informative time series attributes present that can help guide the hyperparameter search. However, the purpose of the synthetic data experiment was to design datasets with a varying amount of information in the temporal and spatial data domains, where it is beneficial to be able to utilize information from both domains. It would therefore defeat the purpose to only use graphs with structures where traditional spatial methods excel, even if they cannot utilize the temporal data. Hence, we desired a good way to adjust the difficulty in distinguishing the different classes from the graph alone, which is somewhat difficult with SBM as the result often is either overly trivial with very dense intra-class communities or too difficult where some or all class communities are near indistinguishable due to very similar intra-class and inter-class edge densities. With BA, it was far easier to adjust the balance to our liking, where some classes are easily distinguishable due to structure and/or edge density, while others may be quite similar to one another. See, for instance, the graphs for the two Balanced datasets in Figure 2 in the paper, where there is a variation in how distinguishable the classes are and using the graph alone is, by intent, not sufficient for an optimal clustering result.
>
> We will attempt to add a brief reasoning for this choice in the revised manuscript.

---

### Author Response · Authors · 2025-07-12
**Revision**

We thank the reviewers for their valuable feedback. Based on the comments received, we have made several changes to improve the paper. In the revised version, the modifications are marked in red. The main changes are summarized as follows.

**Summary of changes**

- Section 2.1 and 5.2: We have clarified the usage of the term spatial and priors and explained how it ties with the graph construction done with CER. This is in response to W2 of reviewer jB4K and their follow-up comments.

- Section 3.1.2: We provided a better explanation for using graph pooling in our framework, which is in response to reviewer jB4K's comments about a lack of proper reasoning/motivation behind its usage.

- Section 3.3: We discussed our choices and assumptions regarding the assessment of the cluster quality, in accordance with our reply to Q3 in reviewer jB4K's initial comments.

- Section 5.1: We provided an explanation about the reason why we use BA to generate graphs in response to the comment of reviewer gKd4.

- Section 5.1: We clarified what $\eta_t$ is in Eq.17 to address the minor comment of reviewer jB4K.

- Section 6.1 and Appendix C.2:  We added new results for models using a graph with static features extracted from the time series, along with a discussion of the results. This is in response to the first listed weakness of reviewer gKd4 regarding lacking comparisons with methods that can process data from both the spatial and temporal domains, and reviewer VMxE’s listed weakness about the lack of comparisons with other STGNNs.

- Section 6.3: We added additional ablation experiments in response to the comments regarding lacking ablations of reviewer jB4K. This includes ablations for the sequence/temporal encoder and the softmax temperature scheduling.

- Section 6.4: We added additional experiments with four variants of our model. The first variant uses a different sequence/temporal encoder in response to a requested ablation of reviewer jB4K. The second variant uses a different message passing layer in the encoder. The third variant introduces a skip connection that bypasses the bottleneck. Finally, the fourth variant uses a higher capacity decoder after the bottleneck. The latter two variants are in response to reviewer VMxE’s comment regarding our findings about the benefit of using a low-parameter decoder.

- Appendix C.2: We modified the discussion regarding the correlation plot obtained for the OnlyGraph setting to clarify that the lack of correlation is expected in this case. This is in response to Q2 in reviewer jB4K’s initial comments.

- Appendix C.3: We added additional plots to show the effect of not using the straight-through trick on the CER dataset. While this was not explicitly requested, this experiment addresses the comments of reviewer jB4K  about the lack of reasoning behind design choices.

---

### Decision · Action_Editor_AwEJ · 2025-08-03

**Recommendation:** Accept as is

**Audience:**

Yes

**Audience Explanation:**

The reviewers agree that there is an audience for this work. I agree with them.

**Claims And Evidence:**

Yes

**Claims Explanation:**

The reviewers agree that there is evidence for the claims in this work.  I agree with them.